# LaplacianFormer: Rethinking Linear Attention with Laplacian Kernel

**Zhe Feng**[2,1] * **Sen Lian**[3] * **Changwei Wang**[5,6][†] **Muyang Zhang**[2,1] **Tianlong Tan**[4] **Rongtao Xu**[7]
**Weiliang Meng**[1,2][†] **Xiaopeng Zhang**[1,2]
[1]MAIS, Institute of Automation, Chinese Academy of Sciences
[2]School of Artificial Intelligence, University of Chinese Academy of Sciences
[3]China Electronics Data Corporation
[4]Institute of Computing Technology, Chinese Academy of Sciences
[5]The Key Laboratory of Computing Power Network and Information Security,
Ministry of Education, Shandong Computer Science Center, Qilu University of Technology
[6]Shandong Provincial Key Laboratory of Computing Power Internet and Service Computing,
Shandong Fundamental Research Center for Computer Science
[7] Spatialtemporal AI
changweiwang@sdas.org, weiliang.meng@ia.ac.cn

## Abstract

The quadratic complexity of softmax attention presents a major obstacle for scaling Transformers to high-resolution vision tasks. Existing linear attention variants often replace the softmax with Gaussian kernels to reduce complexity, but such approximations lack theoretical grounding and tend to oversuppress mid-range token interactions. We propose LaplacianFormer, a Transformer variant that employs a Laplacian kernel as a principled alternative to softmax, motivated by empirical observations and theoretical analysis. To address expressiveness degradation under low-rank approximations, we introduce a provably injective feature map that retains fine-grained token information. For efficient computation, we adopt a Nyström approximation of the kernel matrix and solve the resulting system using Newton–Schulz iteration, avoiding costly matrix inversion and SVD. We further develop custom CUDA implementations for both the kernel and solver, enabling high-throughput forward and backward passes suitable for edge deployment. Experiments on ImageNet show that LaplacianFormer achieves strong performance-efficiency trade-offs while improving attention expressiveness. Code is available at the following site: LaplacianFormer .

## 1 Introduction

The Transformer architecture Vaswani et al. (2017) has become a fundamental framework for sequence modeling, demonstrating strong performance across a wide range of computer vision tasks Jiang et al. (2024); Zhu et al. (2021); Yu et al. (2025); Hou et al. (2024); Su et al. (2024). While its self-attention mechanism effectively captures rich contextual dependencies, its quadratic time and space complexity with respect to sequence length significantly limits scalability to long input sequences Keles et al. (2022); Hassani et al. (2024).

To address this, a number of linear attention variants have been proposed to approximate softmax attention using kernel-based formulations, thereby reducing complexity to linear Katharopoulos et al. (2020); Lu et al. (2021); Chen et al. (2021); Bui et al. (2025); Kashiwagi et al. (2021). Notably, despite differences in implementation, the vast majority of these methods converge on a similar design choice: they rely on Gaussian-like kernels to define attention similarity. This widespread adoption appears to be more of a default convention than a theoretically grounded decision. Indeed, there is a lack of empirical or analytical justification for why the Gaussian kernel is inherently suitable for modeling query-key interactions in attention mechanisms.

---

*Equal contribution.
[†]Corresponding author.

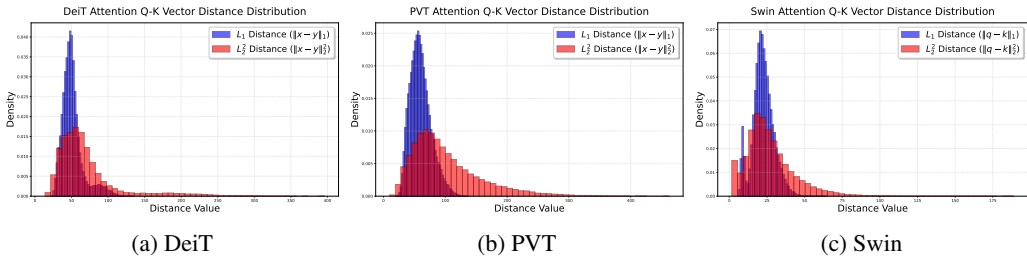

(a) DeiT      (b) PVT      (c) Swin

Figure 1: Distributions of $\ell_1$ and $\ell_2^2$ Q-K distances in DeiT, PVT, and Swin Transformers.

Theoretically, the Gaussian kernel presumes that query-key similarity should decay rapidly with increasing $\ell_2^2$ distance. However, this assumption may not reflect the actual distribution of query-key interactions in vision Transformers. To investigate this issue, we analyze the empirical distribution of query-key distances in DeiT Touvron et al. (2020), PVT Wang et al. (2021b), and Swin Liu et al. (2021b), using official checkpoints on the ImageNet Deng et al. (2009) validation set. As shown in Figure 1, the $\ell_2^2$ distances exhibit a heavy-tailed distribution with high variance and frequent outliers. When passed through the exponential function in the Gaussian kernel, these long-tailed distances will lead to an amplification of the tail effect: outliers dominate the attention map, while moderately relevant keys are overly suppressed. This behavior not only reduces the expressiveness of attention weights but also causes vanishing gradients and unstable optimization, especially during the early stages of training Zhang et al. (2021). In contrast, $\ell_1$ distances tend to be more concentrated and less sensitive to outliers, providing a more faithful measure of token relevance. This observation motivates the use of the Laplacian kernel, defined as $k(x,y) = \exp\left(-\frac{\|x-y\|_1}{\lambda}\right)$, where $x, y \in \mathbb{R}^d$ are input feature vectors and $\lambda > 0$ is a decay parameter. Compared to the Gaussian kernel, which is based on squared $\ell_2$ distances, the Laplacian kernel exhibits a slower decay rate.

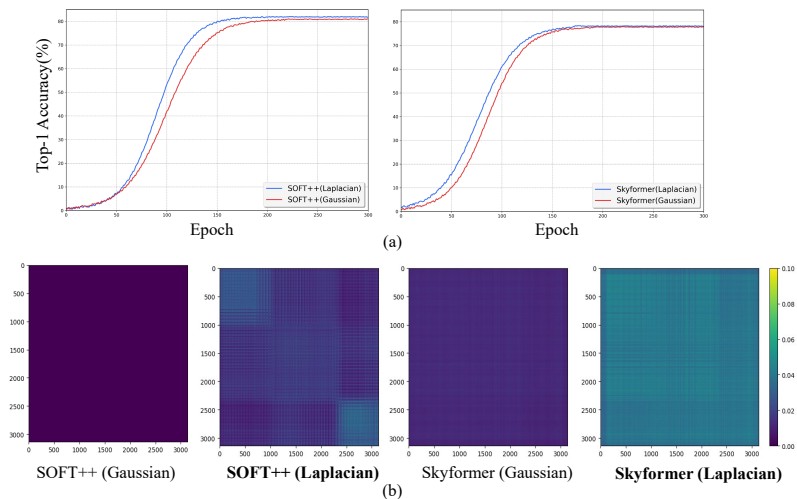

Figure 2: (a) Top-1 accuracy (%) over training epochs on ImageNet. The left plot shows results for SOFT++, and the right plot for Skyformer, each using either a Gaussian or Laplacian kernel for attention computation. Models with Laplacian kernels (blue) converge faster and achieve slightly higher final accuracy compared to their Gaussian counterparts (red). (b) Visual comparison of attention maps with different kernel choices. Each pair shows attention maps from the first Transformer block of SOFT++ and Skyformer, where we only replace the Gaussian kernel with a Laplacian kernel, keeping all other components unchanged. Attention matrices computed with Laplacian kernels exhibit more structured patterns and better-conditioned rank profiles.

Beyond empirical distributions, we further analyze the gradient behavior of these kernels, which plays a critical role in optimization stability. For the Laplacian kernel, the partial derivative with

respect to coordinate $x_i$ is $\frac{\partial k}{\partial x_i} = \frac{1}{\lambda} \cdot \text{sign}(x_i - y_i) \cdot \exp\left(-\frac{\|x-y\|_1}{\lambda}\right)$, while for the Gaussian kernel, it is $\frac{\partial k}{\partial x_i} = \frac{1}{\sigma^2}(x_i - y_i) \cdot \exp\left(-\frac{\|x-y\|_2^2}{2\sigma^2}\right)$, where $\sigma$ denotes the kernel bandwidth. Notably, the Laplacian kernel maintains non-vanishing gradients even when $x$ and $y$ are nearly identical, owing to the piecewise linear nature of the $\ell_1$ norm. In contrast, the Gaussian kernel's gradients diminish linearly as $\|x - y\|_2 \to 0$, resulting in vanishing updates that may hinder convergence. To empirically validate this theoretical claim, we perform a simple ablation by replacing the Gaussian kernel in two representative models—SOFT++ Lu et al. (2024) and Skyformer Chen et al. (2021)—with a Laplacian kernel, keeping all other architectural components unchanged. As shown in Figure 2, this modification alone leads to significantly faster convergence in both models, supporting the hypothesis that the Laplacian kernel facilitates more stable and efficient learning dynamics. Beyond training behavior, we also compare attention maps produced by the two kernels. The same figure also visualizes attention from the first Transformer block of both models. In SOFT++, the Gaussian kernel yields overly sparse attention, while the Laplacian variant produces more expressive and coherent patterns. A similar trend is also observed in Skyformer. These results suggest that, beyond its theoretically favorable decay profile, the Laplacian kernel also improves the practical expressiveness of attention maps, particularly in early to mid-stage layers.

Motivated by these findings, we introduce **LaplacianFormer**, a scalable linear attention framework that replaces the Gaussian-based attention mapping with a Laplacian formulation. To support practical deployment, we develop a CUDA-accelerated implementation that features efficient Laplacian similarity computation and a Newton–Schulz-based inverse solver for fast inference. As shown in

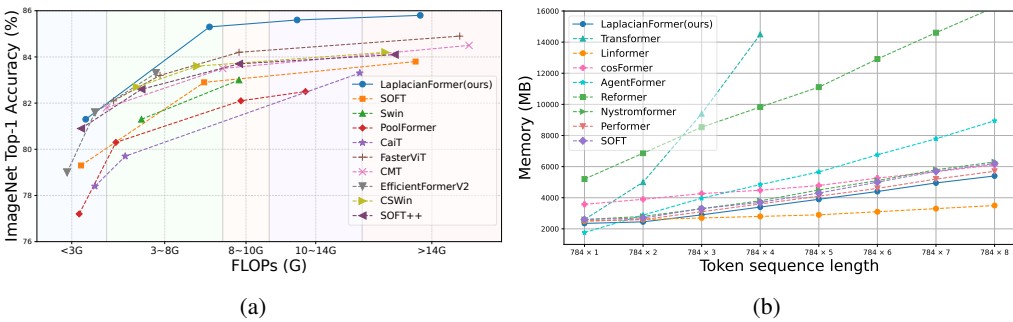

(a)                                                            (b)

Figure 3: Accuracy and Memory Comparison.(a) Top-1 accuracy vs. FLOPs on ImageNet-1k Deng et al. (2009). LaplacianFormer offers a strong accuracy-efficiency trade-off, outperforming prior models across all FLOPs ranges. (b) GPU memory usage across input lengths. LaplacianFormer shows linear scaling, matches efficient Transformers like Performer and SOFT, and far better than the vanilla Transformer.

Figure 3, LaplacianFormer achieves strong performance across accuracy, memory efficiency, and scalability metrics on standard vision benchmarks.

Our main contributions are summarized as follows:

- We propose **LaplacianFormer**, a linear attention model grounded in the Laplacian kernel, which enhances long-range dependency modeling while maintaining scalability and efficiency.
- We develop a CUDA-optimized implementation that integrates Laplacian attention with a Newton–Schulz inverse module, significantly improving runtime and memory efficiency.
- We validate LaplacianFormer on ImageNet-1k Deng et al. (2009) and downstream vision tasks such as object detection and instance segmentation, demonstrating competitive performance across multiple benchmarks.

## 2 RELATED WORK

**Vision Transformer with Softmax Attention.** The Vision Transformer (ViT) Dosovitskiy et al. (2021) has demonstrated exceptional performance and has been widely adopted for a range of

computer vision tasks, including image classification Touvron et al. (2021); Liu et al. (2021b;a); Touvron et al. (2022), object detection Zhu et al. (2021); Zhang et al. (2023a), and semantic segmentation Zheng et al. (2020); Xie et al. (2021); Cheng et al. (2021). By substituting traditional convolutional operations with self-attention mechanisms, ViT enables the modeling of global dependencies within images, offering a powerful alternative to convolutional neural networks (CNNs). However, a major bottleneck lies in the quadratic time and memory complexity, $\mathcal{O}(n^2)$, of standard softmax attention, which significantly restricts its scalability—especially for high-resolution inputs—and limits its deployment on resource-constrained edge devices.

**Linear Attention: A Scalable Alternative.** To mitigate the computational overhead of softmax attention, linear attention has emerged as an efficient alternative. While softmax attention requires $\mathcal{O}(N^2 d)$ operations to compute pairwise similarities, linear attention reduces this to $\mathcal{O}(N d^2)$ by replacing the softmax with kernel-based approximations and reordering computations. Specifically, computing $K^\top V$ first decouples the attention process and enables linear scalability. This efficiency gain becomes particularly significant in modern Transformers, where the token count $N$ typically exceeds the channel dimension $d$. Linear attention thus maintains the ability to model long-range dependencies while substantially improving computational efficiency.

Building on this foundation, a number of linear attention variants have been developed to reduce computational cost while enhancing model capacity. Nyströmformer Xiong et al. (2021) approximates softmax attention via Nyström matrix decomposition. SOFT Lu et al. (2021) replaces softmax with a learnable kernel based on low-rank approximations. Skyformer Chen et al. (2021) incorporates Gaussian kernels and Nyström sampling to improve scalability in vision tasks, while Gaussian Kernelized Attention Kashiwagi et al. (2021) applies a similar design to speech decoding. Performer Choromanski et al. (2021) employs orthogonal random features (FAVOR+) to achieve linear-time softmax approximation. Cosformer Qin et al. (2022) replaces softmax with a cosine-based reweighting scheme to achieve linear complexity. Hedgehog Zhang et al. (2024) introduces structured linear transformations to approximate softmax behavior, providing a unified and scalable alternative. HiViT Zhang et al. (2023b) streamlines hierarchical Transformers by reducing token mixing and applying uniform downsampling.

While differing in architecture, many of these methods share a common reliance on Gaussian kernels to approximate attention weights. In this work, we replace the Gaussian kernel with a Laplace kernel that ensures injectivity and enhances expressiveness, grounded in rigorous theoretical analysis.

## 3 PRELIMINARIES

### 3.1 SOFTMAX SELF-ATTENTION

Softmax self-attention is a core operation in transformer models. Given an input sequence $\mathbf{X} \in \mathbb{R}^{N \times d_e}$ of $N$ tokens embedded in a $d_e$-dimensional space, we compute queries, keys, and values via linear projections: $\mathbf{Q} = \mathbf{X}\mathbf{W}_Q, \mathbf{K} = \mathbf{X}\mathbf{W}_K, \mathbf{V} = \mathbf{X}\mathbf{W}_V$, where $\mathbf{W}_Q, \mathbf{W}_K, \mathbf{W}_V \in \mathbb{R}^{d_e \times d}$ are learnable parameters and $\mathbf{Q}, \mathbf{K}, \mathbf{V} \in \mathbb{R}^{N \times d}$. The standard scaled dot-product attention for token $i$ is:

$$\text{Attention}(\mathbf{Q}, \mathbf{K}, \mathbf{V})_i = \frac{\sum_{j=1}^{N} \exp\left(\frac{\mathbf{q}_i^\top \mathbf{k}_j}{\sqrt{d}}\right) \mathbf{v}_j}{\sum_{j=1}^{N} \exp\left(\frac{\mathbf{q}_i^\top \mathbf{k}_j}{\sqrt{d}}\right)} \tag{1}$$

and in matrix form:

$$\text{Attention}(\mathbf{Q}, \mathbf{K}, \mathbf{V}) = \text{Softmax}\left(\frac{\mathbf{Q}\mathbf{K}^\top}{\sqrt{d}}\right) \mathbf{V} \tag{2}$$

This formulation computes a similarity matrix $\mathbf{Q}\mathbf{K}^\top \in \mathbb{R}^{N \times N}$, resulting in $\mathcal{O}(N^2 d)$ complexity due to all pairwise interactions. To reduce cost, consider removing the softmax. Without it, attention simplifies to $\left(\frac{\mathbf{Q}\mathbf{K}^\top}{\sqrt{d}}\right) \mathbf{V}$, which can be reordered as $\mathbf{Q}(\mathbf{K}^\top \mathbf{V})$ using associativity. This avoids forming the large $N \times N$ matrix and reduces complexity to $\mathcal{O}(N d^2)$, linear in $N$ if $d$ is small. This insight underlies linear attention, which replaces softmax with associative operations for improved efficiency. Figure 4 compares softmax and linear self-attention.

Figure 4: Comparison between Softmax Self-Attention (left) and Linear Self-Attention (right). The former computes a full $N \times N$ similarity matrix, while the latter enables associativity through kernel decomposition, reducing the complexity from $\mathcal{O}(N^2)$ to $\mathcal{O}(N)$.

## 3.2 LINEAR SELF-ATTENTION

Linear self-attention reformulates the attention mechanism by approximating the similarity computation through kernel-based feature mappings. Specifically, let $\phi(\cdot)$ denote a kernel function, and define the similarity between a query $\mathbf{q}_i$ and a key $\mathbf{k}_j$ as: $\mathrm{Sim}(\mathbf{q}_i, \mathbf{k}_j) = \phi(\mathbf{q}_i)\,\phi(\mathbf{k}_j)^\top$.

The kernel-based formulation replaces the exponential dot product with a more general similarity function, enabling efficient reordering of computations and eliminating the softmax operation. The attention output for the $i$-th query can be written as:

$$\mathrm{Attention}(\mathbf{Q}, \mathbf{K}, \mathbf{V})_i = \frac{\phi(\mathbf{q}_i)\left(\sum_{j=1}^{N} \phi(\mathbf{k}_j)^\top \mathbf{v}_j\right)}{\phi(\mathbf{q}_i)\left(\sum_{j=1}^{N} \phi(\mathbf{k}_j)^\top\right)}. \tag{3}$$

Since the key-value summaries $\sum_{j=1}^{N} \phi(\mathbf{k}_j)^\top \mathbf{v}_j$ and $\sum_{j=1}^{N} \phi(\mathbf{k}_j)^\top$ are independent of the query, they can be precomputed, allowing each attention output to be computed in linear time.

## 4 METHOD

### 4.1 LAPLACIANFORMER

Our LaplacianFormer instantiates the general kernel attention framework described in Section 3.2 using a novel Laplace-based transformation inspired by recent work on attention injectivity Han et al. (2024a). Instead of directly using the Laplacian kernel as a similarity score, we construct a normalized kernel representation for each query $\mathbf{q}_i$ to enhance feature discrimination:

$$\mathbf{z}_i = \mathbf{\Sigma}^{-\frac{1}{2}} \left( [k(\mathbf{q}_i, \mathbf{k}_1), \ldots, k(\mathbf{q}_i, \mathbf{k}_N)]^\top - \frac{1}{N} \sum_{j=1}^{N} k(\mathbf{q}_i, \mathbf{k}_j) \right) + \frac{1}{N}, \tag{4}$$

where $k(\mathbf{q}, \mathbf{k}) = \exp\left(-\frac{\|\mathbf{q} - \mathbf{k}\|_1}{\lambda}\right)$ denotes the Laplacian kernel. The whitening matrix $\mathbf{\Sigma}^{-1/2} \in \mathbb{R}^{N \times N}$ is ideally computed from the covariance of query–key similarity vectors $\mathbf{g}_i \in \mathbb{R}^N$, where each $\mathbf{g}_i = [k(\mathbf{q}_i, \mathbf{k}_1), \ldots, k(\mathbf{q}_i, \mathbf{k}_N)]^\top$.

In practice, computing the full inverse square root $\mathbf{\Sigma}^{-1/2}$ is computationally prohibitive for long sequences, requiring eigendecomposition with $\mathcal{O}(N^3)$ time and $\mathcal{O}(N^2)$ memory. To mitigate this, we approximate the whitening operation with a diagonal estimator that normalizes each feature dimension independently across a batch of query–key vectors $\{\mathbf{g}_i\}_{i=1}^{B}$, where $B$ is the batch size.

For each dimension $j \in \{1, \ldots, N\}$, we compute the empirical mean and variance:

$$\mu_j = \frac{1}{B} \sum_{i=1}^{B} \mathbf{g}_{ij}, \quad \sigma_j^2 = \frac{1}{B} \sum_{i=1}^{B} (\mathbf{g}_{ij} - \mu_j)^2. \tag{5}$$

We then normalize each element of the similarity vector: $\tilde{\mathbf{g}}_{ij} = \frac{\mathbf{g}_{ij} - \mu_j}{\sqrt{\sigma_j^2 + \varepsilon}}$, where $\varepsilon$ is a small constant added for numerical stability. This corresponds to a diagonal whitening matrix:

$$\mathbf{D}^{-1/2} = \mathrm{diag}\left(\frac{1}{\sqrt{\sigma_1^2 + \varepsilon}}, \ldots, \frac{1}{\sqrt{\sigma_N^2 + \varepsilon}}\right). \tag{6}$$

This approximation preserves the centering and scaling effects of full whitening, improves stability, and reduces both time and memory complexity from quadratic to linear in $N$, making it compatible with efficient kernelized attention. For completeness, we define the kernel similarity matrix among keys as $\mathbf{G}_{ij} = k(\mathbf{k}_i, \mathbf{k}_j)$, $\mathbf{\Sigma}_{\text{key}} = \mathbf{P}\mathbf{G}\mathbf{P}^\top$, with $\mathbf{P} = \mathbf{I} - \frac{1}{N}\mathbf{1}\mathbf{1}^\top$. Although not directly used in normalization, the key–key covariance $\mathbf{\Sigma}_{\text{key}}$ provides a useful interpretation of the kernel structure.

We prove in the appendix that the transformation in Eq. equation 4 is injective under mild assumptions, ensuring that distinct queries yield distinct outputs. This injectivity property aligns with the behavior of softmax attention, which is inherently injective and yields full-rank attention maps that preserve fine-grained token distinctions Han et al. (2024a). The final attention output incorporates both global interactions via kernelized similarity and local context modeling through depth-wise convolution. Specifically, we compute

$$\text{Attention}(\mathbf{Q}, \mathbf{K}, \mathbf{V}) = \mathbf{Z}\mathbf{V} + \text{DWC}(\mathbf{V}), \tag{7}$$

where $\mathbf{Z} \in \mathbb{R}^{N \times N}$ stacks each $\mathbf{z}_i^\top$ as a row, and $\text{DWC}(\cdot)$ denotes a depth-wise convolution applied over the value sequence $\mathbf{V}$.

## 4.2 Nyström Approximation for Laplacian Kernel

To efficiently compute Laplacian kernel-based attention, we adopt a Nyström approximation Williams & Seeger (2000); Xiong et al. (2021). The Nyström method approximates the kernel matrix $\mathbf{G}$ by selecting a small set of landmark keys and computing a rank-reduced estimate $\widetilde{\mathbf{G}} \in \mathbb{R}^{N \times N}$, defined as $\widetilde{\mathbf{G}} = \mathbf{C}\mathbf{W}^\dagger\mathbf{C}^\top$, where $\mathbf{C} \in \mathbb{R}^{N \times m}$ is the matrix of Laplacian kernel similarities between all queries and a selected subset of $m \ll N$ landmark keys, $\mathbf{W} \in \mathbb{R}^{m \times m}$ contains pairwise Laplacian kernel similarities among the $m$ selected landmark keys, and $\mathbf{W}^\dagger$ denotes the Moore–Penrose pseudoinverse of $\mathbf{W}$. More specifically, the $(i, \ell)$-th entry of $\mathbf{C}$ is defined as:

$$\mathbf{C}_{i\ell} = k(\mathbf{q}_i, \tilde{\mathbf{k}}_\ell) = \exp\left(-\frac{1}{\lambda}\left\|\mathbf{q}_i - \tilde{\mathbf{k}}_\ell\right\|_1\right), \tag{8}$$

where $\mathbf{q}_i$ is the query vector of the $i$-th token and $\tilde{\mathbf{k}}_\ell \in \{\mathbf{k}_1, \ldots, \mathbf{k}_N\}$ is the $\ell$-th landmark key, while the $(\ell, \ell')$-th entry of $\mathbf{W}$ is computed as:

$$\mathbf{W}_{\ell\ell'} = k(\tilde{\mathbf{k}}_\ell, \tilde{\mathbf{q}}_{\ell'}) = \exp\left(-\frac{1}{\lambda}\left\|\tilde{\mathbf{k}}_\ell - \tilde{\mathbf{q}}_{\ell'}\right\|_1\right), \tag{9}$$

where $\tilde{\mathbf{k}}_\ell$ and $\tilde{\mathbf{q}}_{\ell'}$ are the landmark key and query vectors selected by Nyström sampling, respectively.

The process for computing the low-rank Laplacian kernel via Nyström approximation is outlined in Algorithm 1. In Line 2, the sampling function $f_s$ selects $m \ll N$ landmark tokens from the full set of queries and keys, forming the landmark matrices $\widetilde{\mathbf{Q}}, \widetilde{\mathbf{K}} \in \mathbb{R}^{m \times d}$. Lines 3–5 perform the core kernel operations: Line 3 computes the landmark kernel matrix $\mathbf{W}$ (Eq. 9), Line 4 computes the query-to-landmark kernel matrix $\mathbf{C}$ (Eq. 8), and Line 5 applies the Nyström approximation using $\mathbf{W}^\dagger$ to obtain the final attention matrix $\hat{\mathbf{S}}$.

---

**Algorithm 1** Laplacian Kernel with Nyström Approximation

---

1: **Input:** Queries $Q \in \mathbb{R}^{N \times d}$, Keys $K \in \mathbb{R}^{N \times d}$, Nyström sampling function $f_s$
2: **Sampling:** $\tilde{Q}, \tilde{K} \leftarrow f_s(Q), f_s(K)$               ▷ Select $m \ll n$ landmark points
3: $W \leftarrow \exp\left(-\frac{1}{\lambda}\|\tilde{Q} \ominus \tilde{K}\|_1\right)$              ▷ Kernel matrix on sampled points
4: $C \leftarrow \exp\left(-\frac{1}{\lambda}\|Q \ominus \tilde{K}\|_1\right)$        ▷ Cross-kernel between all queries and landmarks
5: $\hat{G} \leftarrow CW^\dagger C^\top$             ▷ Low-rank approximation of full kernel matrix
6: **Output:** $\hat{G}$

---

**Laplacian Kernel Inversion via Newton–Schulz Iteration.** To efficiently and stably approximate the inverse of the landmark kernel matrix $\mathbf{W} \in \mathbb{R}^{m \times m}$, which is symmetric and positive semi-definite, we use the Newton–Schulz iteration. Since convergence requires $\mathbf{W}$ to be strictly positive

---

**Algorithm 2** Newton–Schulz Iteration for Approximating $\mathbf{W}^\dagger$

---

1: **Input:** Landmark kernel matrix $\mathbf{W} \in \mathbb{R}^{m \times m}$, number of iterations $\mathcal{T} \in \mathbb{Z}^+$
2: Add small perturbation: $\mathbf{W} \leftarrow \mathbf{W} + \epsilon \mathbf{I}$, where $\epsilon > 0$
3: Initialize scaling factor: $\boldsymbol{\alpha} \leftarrow \frac{2}{\|\mathbf{W}\|_2}$
4: Initialize: $\mathbf{X}_0 \leftarrow \boldsymbol{\alpha} \mathbf{W}^\top$
5: **for** $k = 1$ to $\mathcal{T}$ **do**
6: $\quad \mathbf{X}_k \leftarrow \mathbf{X}_{k-1}(2\mathbf{I} - \mathbf{W}\mathbf{X}_{k-1})$
7: **end for**
8: **Output:** Approximate pseudoinverse $\mathbf{X}_{\mathcal{T}} \approx \mathbf{W}^\dagger$

---

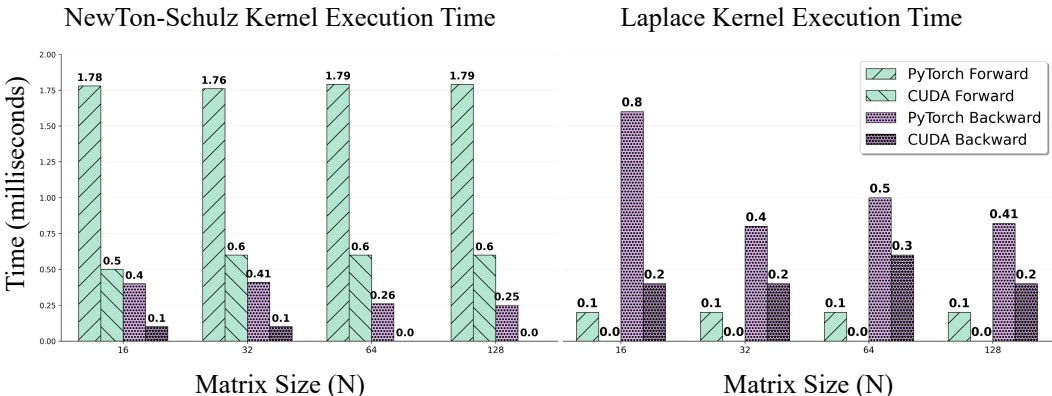

Figure 5: **Execution time breakdown of custom CUDA kernels.** Comparison of forward and backward execution time for Newton–Schulz iteration (left) and Laplacian kernel (right) across different matrix sizes (batch = 1, 2 heads, 32 channels). CUDA execution times ($< 0.05$ms) are shown as 0.0 due to timing resolution limits.

definite, we apply a small diagonal perturbation $\mathbf{W} \leftarrow \mathbf{W} + \epsilon \mathbf{I}$, with $\epsilon > 0$, preserving the structure while ensuring stability. Unlike inversion or SVD-based methods, Newton–Schulz relies only on matrix multiplications and additions, making it GPU-friendly. The iteration starts with $\mathbf{X}_0 = \boldsymbol{\alpha} \mathbf{W}^\top$, where $\boldsymbol{\alpha} = \frac{2}{\|\mathbf{W}\|_2}$ ensures $\|\mathbf{I} - \boldsymbol{\alpha} \mathbf{W}\mathbf{W}^\top\| < 1$. The update rule is: $\mathbf{X}_{k+1} = \mathbf{X}_k(2\mathbf{I} - \mathbf{W}\mathbf{X}_k)$. The full algorithm is detailed in Algorithm 2.

**Sampling Strategies for Landmark Selection.** To efficiently approximate attention, we adopt a pooling-based landmark selection strategy inspired by PVTv2 Wang et al. (2021a). The query tensor $\mathbf{Q} \in \mathbb{R}^{B \times H \times N \times d}$ is reshaped into a spatial map $\mathbb{R}^{B \cdot H \times d \times H' \times W'}$, where $N = H' \times W'$. We apply average pooling with kernel size $r$ and stride $r$ to aggregate each $r \times r$ region into a landmark token, yielding $\frac{H'}{r} \times \frac{W'}{r}$ tokens per head.

We also explored a depthwise convolution-based selection strategy, in which each $r \times r$ region is processed by a lightweight filter to extract local structure. While this approach offers greater expressiveness, it yielded no significant improvement over average pooling in our experiments. Given its higher computational cost and additional parameters, we adopt average pooling by default.

**Convergence Guarantee and Complexity Analysis.** The Newton–Schulz iteration is guaranteed to converge for strictly positive definite matrices; this condition is satisfied by applying a small diagonal perturbation to $\mathbf{W}$. Our method achieves linear time and space complexity $\mathcal{O}(n)$ with respect to the input length $n$. A complete complexity analysis and proof of convergence are provided in the appendix.

## 4.3 CUDA ACCELERATION

The Laplacian kernel fuses distance computation and exponential transformation into a single operation, reducing global memory access. For Newton–Schulz iteration, we optimize matrix multiplications via tiling and register reuse.

Table 1: Performance comparison with state-of-the-art models on ImageNet.

| FLOPs range | Model | Params | FLOPs | Top-1 %↑ | Image Size |
|---|---|---|---|---|---|
| < 3G | Agent-Deit-T Han et al. (2024c) | 6.0M | 1.2G | 74.9 | 224 |
| | VRWKV-T Duan et al. (2025) | 6.2M | 1.2G | 75.1 | 256 |
| | PVT-T-PolaFormer Meng et al. (2025) | 12M | 2.0G | 78.8 | 224 |
| | FL-PVTv2-B1 Han et al. (2023) | 13M | 2.2G | 79.5 | 224 |
| | BiFormer-T Zhu et al. (2023) | 13.1M | 2.2G | 81.4 | 224 |
| | LaplacianFormer-Tiny | 12.1M | 2.1G | **81.4** | 224 |
| 3∼8G | InLine-CSwin-S Han et al. (2024a) | 33M | 6.8G | 83.8 | 224 |
| | SViT-S Huang et al. (2023) | 25M | 4.4G | 83.6 | 224 |
| | BiFormer-S Zhu et al. (2023) | 25.5M | 4.5G | 83.8 | 224 |
| | HiViT-T Zhang et al. (2023b) | 19M | 4.6G | 82.1 | 224 |
| | Agent-PVT-S Han et al. (2024c) | 20.6M | 4.0G | 82.2 | 224 |
| | LaplacianFormer-Small | 25.7M | 4.8G | **83.8** | 224 |
| 8∼10G | SViT-B Huang et al. (2023) | 52M | 9.9G | 84.8 | 224 |
| | SOFT++-Medium Lu et al. (2024) | 45M | 7.2G | 83.7 | 224 |
| | BiFormer-B Zhu et al. (2023) | 56.8M | 9.8G | 84.3 | 224 |
| | Swin-S-PolaFormer Meng et al. (2025) | 50M | 8.7G | 83.6 | 224 |
| | SLAB-Swin-S Guo et al. (2024) | - | 8.7G | 81.8 | 224 |
| | LaplacianFormer-Medium | 46.3M | 7.43G | **85.3** | 224 |
| 10∼14G | StructViT-B-8-1 Kim et al. (2024) | 52M | 12G | 84.3 | 224 |
| | SOFT++-Large Lu et al. (2024) | 64M | 11G | 84.1 | 224 |
| | NAT-B Hassani et al. (2023) | 90M | 13.7G | 84.3 | 224 |
| | MogaNet-L Li et al. (2024) | 82.5M | 15.9G | 84.7 | 224 |
| | FLatten-CSwin-S Han et al. (2023) | 35M | 6.9G | 83.6 | 224 |
| | LaplacianFormer-Large | 63.1M | 11.2G | **85.6** | 224 |
| >14G | VRWKV-B Duan et al. (2025) | 93.7M | 18.2G | 82.0 | 224 |
| | SViT-L Huang et al. (2023) | 95M | 15.6G | 85.3 | 224 |
| | MLLA-B Han et al. (2024b) | 96M | 16.2G | 85.3 | 224 |
| | HiViT-B Zhang et al. (2023b) | 66M | 15.9G | 83.8 | 224 |
| | LaplacianFormer-Huge | 78.5M | 15.5G | **85.8** | 224 |

To assess the effectiveness of our CUDA acceleration, we compare the execution time of both Laplacian kernel evaluation and Newton–Schulz iteration against their PyTorch counterparts Paszke et al. (2017), with and without custom CUDA kernels. As shown in Figure 5, our implementation consistently outperforms the baseline across various matrix sizes. The speedup is particularly prominent in backward passes, which benefit from precomputed gradients and in-place memory reuse. Numerical accuracy comparisons are provided in the appendix.

## 5 EXPERIMENTS

### 5.1 IMAGE CLASSIFICATION

**Datasets and model architectures.** We evaluate our model on the ImageNet-1K dataset Deng et al. (2009), which contains 1.28M training and 50K validation images across 1000 classes. Built on the PVT architecture Wang et al. (2021b), our LaplacianFormer re-designs the self-attention mechanism by constructing an injective attention function based on the Laplacian kernel. To ensure training efficiency, we implement two custom CUDA kernels: one for computing the Laplacian kernel matrix and another for performing Newton–Schulz iteration to approximate the inverse. Additionally, RoPE SU2 (2024) is adopted for positional encoding. All other settings follow the original PVT configuration. Training is performed with a batch size of 1024 on multiple NVIDIA H800 GPUs.

**Comparison.** We compare the Top-1 accuracy and computational cost of our LaplacianFormer against state-of-the-art Vision Transformers. As shown in Table 1, models are grouped by FLOPs:<1G, 1–3G, 3–5G, 5–10G, and >10G. LaplacianFormer consistently achieves the highest Top-1 accuracy across all FLOP ranges. This result demonstrates the superiority of LaplacianFormer over existing methods.

### 5.2 OBJECT DETECTION AND INSTANCE SEGMENTATION

**Results.** Table 2 summarizes the comparison results under the $1\times$ schedule for both Mask R-CNN He et al. (2017) and RetinaNet Lin et al. (2017). Across all scales, LaplacianFormer consistently surpasses previous backbone designs. For instance, LaplacianFormer-Tiny achieves 43.2 $AP^b$

and 40.3 $AP^m$ under Mask R-CNN, outperforming SOFT++-Tiny and FL-PVT-T. Under RetinaNet, it further achieves 42.5 $AP^b$, ranking first among all tiny-scale counterparts. As the model size increases, LaplacianFormer-Medium yields 48.0 $AP^b$ and 43.5 $AP^m$, establishing a new state-of-the-art within the medium-sized category. These results highlight the strong generalization and detection capabilities enabled by our Laplacian kernel attention mechanism.

Table 2: Comparison to other backbones using RetinaNet and Mask R-CNN with "1×" schedule.

| Backbone | Mask R-CNN 1× | | | | | | RetinaNet 1× | | | | | |
| --- | --- | --- | --- | --- | --- | --- | --- | --- | --- | --- | --- | --- |
| | $AP^b$ | $AP^b_{50}$ | $AP^b_{75}$ | $AP^m$ | $AP^m_{50}$ | $AP^m_{75}$ | $AP^b$ | $AP^b_{50}$ | $AP^b_{75}$ | $AP^b_S$ | $AP^b_M$ | $AP^b_L$ |
| Swin-T-PRepBN Guo et al. (2024) | 42.9 | 65.8 | 46.8 | 39.3 | 62.6 | 41.9 | – | – | – | – | – | – |
| FL-PVT-T Han et al. (2023) | 38.2 | 61.6 | 41.9 | 37.0 | 57.6 | 39.0 | – | – | – | – | – | – |
| SOFT++-Tiny Lu et al. (2024) | 41.2 | 63.7 | 44.7 | 38.2 | 61.0 | 41.0 | 41.9 | 62.7 | 44.7 | 27.8 | 45.4 | 55.6 |
| LaplacianFormer-Tiny | **43.2** | **66.1** | **47.2** | **40.3** | **63.0** | **42.9** | **42.5** | **64.1** | **46.4** | **29.1** | **46.9** | **57.8** |
| PVT-S-PolaFormer Meng et al. (2025) | 43.9 | 66.1 | 47.9 | 40.2 | 63.1 | 43.0 | 43.2 | 64.1 | 46.4 | – | – | – |
| InLine-PVT-S Han et al. (2024a) | 43.4 | 66.4 | 47.1 | 40.1 | 63.1 | 43.3 | – | – | – | – | – | – |
| SOFT++-Small Lu et al. (2024) | 43.8 | 66.0 | 47.5 | 40.1 | 63.0 | 43.0 | 43.7 | 64.9 | 46.8 | 28.7 | 47.4 | 57.6 |
| LaplacianFormer-Small | **45.8** | **68.2** | **49.8** | **42.0** | **65.1** | **45.2** | **45.5** | **66.8** | **49.1** | **30.7** | **51.8** | **59.5** |
| Agent-PVT-M Han et al. (2024c) | 45.9 | 67.8 | 50.4 | 42.0 | 65.0 | 45.4 | – | – | – | – | – | – |
| FL-Swin-M Han et al. (2023) | 44.0 | 66.4 | 48.0 | 40.3 | 63.4 | 43.5 | – | – | – | – | – | – |
| SOFT++-Medium Lu et al. (2024) | 46.6 | 67.8 | 51.2 | 42.0 | 64.8 | 45.2 | 44.3 | 64.7 | 47.4 | 29.0 | 48.2 | 59.9 |
| LaplacianFormer-Medium | **48.0** | **70.3** | **52.5** | **43.5** | **65.8** | **46.5** | **47.2** | **68.5** | **51.5** | **31.8** | **53.0** | **61.4** |
| Swin-T-PolaFormer Meng et al. (2025) | 44.8 | 67.6 | 49.1 | 40.5 | 64.1 | 43.5 | – | – | – | – | – | – |
| Agent-PVT-L Han et al. (2024c) | 46.9 | 69.2 | 51.4 | 42.8 | 66.2 | 46.2 | – | – | – | – | – | – |
| SOFT++-Large Lu et al. (2024) | 47.0 | 68.3 | 51.7 | 42.2 | 65.2 | 45.4 | 47.0 | 67.8 | 50.4 | 30.2 | 50.9 | 62.0 |
| LaplacianFormer-Large | **48.2** | **70.5** | **53.0** | **43.8** | **67.1** | **47.4** | **48.5** | **69.3** | **52.4** | **32.6** | **52.3** | **63.8** |

## 5.3 Ablation Studies

**Convergence Under Varying Condition Numbers.** We evaluate solver convergence across varying condition numbers. We measure the relative error for Newton–Schulz using the Frobenius norm $\|X_k - W^\dagger\|_F / \|W^\dagger\|_F$, and for CG using the Euclidean norm $\|x_k - x^*\|_2 / \|x^*\|_2$. As shown in Figure 6, CG converges more rapidly under well-conditioned settings (e.g., $\kappa = 2$) but degrades significantly as the condition number increases. In contrast, Newton–Schulz exhibits an initial warm-up phase followed by stable convergence even under ill-conditioned regimes (e.g., $\kappa = 50$), indicating greater robustness in practice.

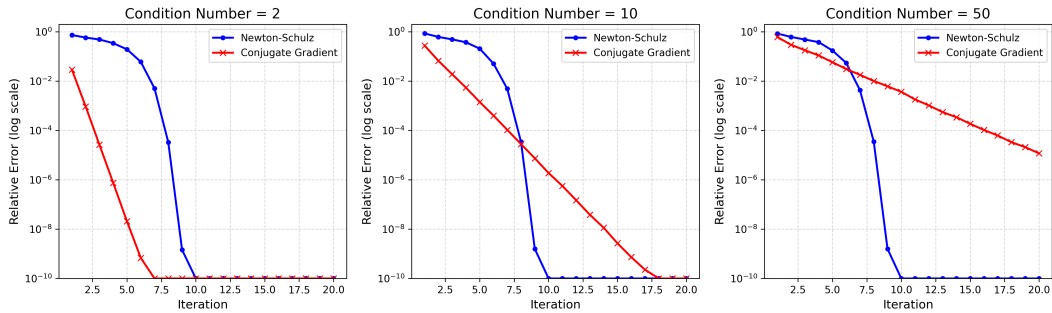

Figure 6: Convergence behavior of Newton–Schulz and conjugate gradient methods under varying condition numbers. Each plot shows relative error (log scale) vs. iteration.

**Inverse Solver Effect.** We compare two iterative solvers—Newton–Schulz (used in our model) and conjugate gradient (CG)—for computing the kernel inverse in linear attention, both implemented with custom CUDA kernels. As shown in Table 3, Newton–Schulz achieves higher Top-1 accuracy than CG for both LaplacianFormer-Tiny (81.4% vs. 79.2%) and LaplacianFormer-Small (83.8% vs. 81.4%), likely due to better GPU convergence and numerical stability.

**Effect of Laplacian Kernel Scale.** We study the impact of the Laplacian kernel scale $\lambda$ in the similarity function $\text{sim}_{\text{Lap}}(q, k) = \exp\left(-\frac{\|q-k\|_1}{\lambda}\right)$. As shown in Table 3, the model achieves the

best Top-1 accuracy (81.4%) when $\lambda = 4$. Small $\lambda$ values (e.g., 0.5, 1) overly suppress long-range interactions, while large values (e.g., 8) yield overly smooth attention, diluting local detail. An intermediate scale ($\lambda = 4$) balances local sensitivity and global context, and is thus fixed in all experiments. Attention map visualizations (Figure 7) further validate this choice.

Table 3: **Ablation studies on LaplacianFormer architecture.** *(left)* Top-1 accuracy (%) of LaplacianFormer variants using different inverse solvers: conjugate gradient (CG) vs. Newton–Schulz (NS). *(right)* Effects of the Laplacian kernel scale $\lambda$ on LaplacianFormer-Tiny.

| Model | CG (%) | NS (%) |
|---|---|---|
| LaplacianFormer-Tiny | 79.2 | **81.4** |
| LaplacianFormer-Small | 81.4 | **83.8** |

| $\lambda$ | 0.5 | 1 | 2 | 4 | 8 |
|---|---|---|---|---|---|
| Top-1 Acc (%) ↑ | 79.4 | 79.6 | 80.1 | **81.4** | 78.5 |

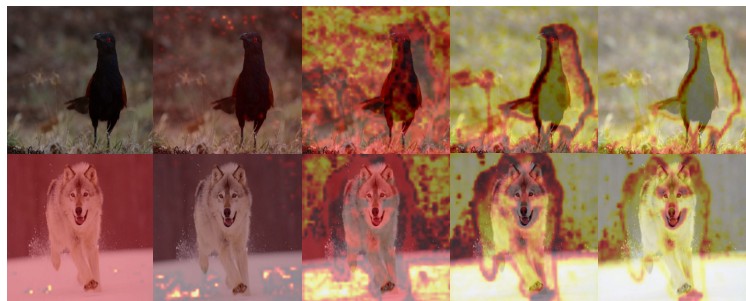

Figure 7: **Visualization of attention maps under different Laplacian kernel scales $\lambda$.** From left to right: $\lambda = 0.5, 1, 2, 4, 8$.

## 6 CONCLUSIONS AND FUTURE WORK

We propose **LaplacianFormer**, a Transformer variant that employs a Laplacian kernel to construct injective and normalized attention, enabling fine-grained token discrimination with linear complexity. To ensure scalability, we adopt the Nyström approximation and accelerate computation via Newton–Schulz iteration, with efficient CUDA support for both forward and backward passes. LaplacianFormer strikes a balance between expressiveness and efficiency, performing well on both vision and long-sequence tasks. Moreover, it achieves strong results on downstream applications such as object detection and segmentation, further demonstrating its generalization capability.

This work specifically focuses on comparing Laplacian and Gaussian kernels—the latter being the dominant choice in prior linear attention models Katharopoulos et al. (2020); Lu et al. (2021); Chen et al. (2021). Our goal is to challenge this convention through both theoretical analysis and empirical validation. Broader comparisons with other kernel families (e.g., cosine, polynomial) are left as future work.

### ACKNOWLEDGMENTS

This work was supported by Beijing Natural Science Foundation (L231013, L241056, JQ23014), National Natural Science Foundation of China (Nos. 62376271, U22B2034, 62572059, and 62365014), Jiangxi Provincial Natural Science Foundation (No. 20253BAC280104), Shenzhen S&T Programme (No. CJGJZD20240729141906008), Taishan Scholars Program No.TSQN202507241, Key R&D Program of Shandong Province, China No.2025KJHZ013, Shandong Provincial University Youth Innovation and Technology Support Program No.2022KJ291, Shandong Provincial Natural Science Foundation for Young Scholars Program No.ZR2025QC1627, Qilu University of Technology (Shandong Academy of Sciences) Youth Outstanding Talent Program No. 2024QZJH02, Open Project of Key Laboratory of Computing Power Network and Information Security (No. 2024PY021), and the Open Project Program of State Key Laboratory of Virtual Reality Technology and Systems, Beihang University (No. VRLAB2025B03).

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

## A  GRADIENT BEHAVIOR OF GAUSSIAN VS. LAPLACIAN KERNELS

**Proposition 1** (Gradient Magnitude Decay in High Dimensions). *Let $\mathbf{x}, \mathbf{y} \in \mathbb{R}^d$. As the distance $\|\mathbf{x} - \mathbf{y}\| \to 0$, the gradient magnitude of the Gaussian kernel decays linearly to zero. In contrast, the gradient magnitude of the Laplacian kernel remains at a constant non-zero order proportional to $\sqrt{d}$.*

*Proof.* Let $\mathbf{t} = \mathbf{x} - \mathbf{y} \in \mathbb{R}^d$. We analyze the $\ell_2$ norm of the gradient with respect to $\mathbf{x}$, denoted as $\|\nabla_{\mathbf{x}} k(\mathbf{x}, \mathbf{y})\|_2$.

**1. Gaussian kernel**
The multi-dimensional Gaussian kernel is defined as:

$$k_{\text{Gauss}}(\mathbf{x}, \mathbf{y}) = \exp\left(-\frac{\|\mathbf{t}\|_2^2}{2\sigma^2}\right)$$

The gradient vector with respect to $\mathbf{x}$ is:

$$\nabla_{\mathbf{x}} k_{\text{Gauss}} = -\frac{\mathbf{t}}{\sigma^2} \exp\left(-\frac{\|\mathbf{t}\|_2^2}{2\sigma^2}\right)$$

Taking the $\ell_2$ norm of the gradient yields:

$$\|\nabla_{\mathbf{x}} k_{\text{Gauss}}\|_2 = \frac{\|\mathbf{t}\|_2}{\sigma^2} \exp\left(-\frac{\|\mathbf{t}\|_2^2}{2\sigma^2}\right)$$

As $\|\mathbf{t}\|_2 \to 0$, the exponential term approaches 1, resulting in:

$$\|\nabla_{\mathbf{x}} k_{\text{Gauss}}\|_2 \sim \frac{\|\mathbf{t}\|_2}{\sigma^2}$$

This confirms that the gradient magnitude of the Gaussian kernel diminishes **linearly** to zero as $\mathbf{x} \to \mathbf{y}$.

**2. Laplacian kernel**
The multi-dimensional Laplacian kernel is defined as:

$$k_{\text{Laplace}}(\mathbf{x}, \mathbf{y}) = \exp\left(-\frac{\|\mathbf{t}\|_1}{\lambda}\right)$$

For $t_i \neq 0$ ($\forall i \in \{1, \ldots, d\}$), the gradient vector is:

$$\nabla_{\mathbf{x}} k_{\text{Laplace}} = -\frac{1}{\lambda} \text{sign}(\mathbf{t}) \exp\left(-\frac{\|\mathbf{t}\|_1}{\lambda}\right)$$

where $\text{sign}(\mathbf{t}) \in \{-1, 1\}^d$ is applied element-wise. The $\ell_2$ norm of this sign vector is $\|\text{sign}(\mathbf{t})\|_2 = \sqrt{d}$. Taking the $\ell_2$ norm of the gradient yields:

$$\|\nabla_{\mathbf{x}} k_{\text{Laplace}}\|_2 = \frac{\sqrt{d}}{\lambda} \exp\left(-\frac{\|\mathbf{t}\|_1}{\lambda}\right)$$

As $\mathbf{t} \to \mathbf{0}$, the exponential term approaches 1, resulting in:

$$\|\nabla_{\mathbf{x}} k_{\text{Laplace}}\|_2 \sim \frac{\sqrt{d}}{\lambda}$$

Thus, in a $d$-dimensional space, the gradient magnitude of the Laplacian kernel is bounded away from zero and remains at a **constant order** $\sqrt{d}/\lambda$. $\qquad\square$

## B  MATHEMATICAL PROOF

This section offers mathematical proofs for the propositions outlined in the main paper.

**Proposition 2** (Injectivity of Laplacian-based Kernel Embedding). *Let $\mathbf{q}_i, \mathbf{q}_j \in \mathbb{R}^d$, and define their kernel similarity vectors as*

$$\mathbf{g}_i = [k(\mathbf{q}_i, \mathbf{k}_1), \dots, k(\mathbf{q}_i, \mathbf{k}_N)]^\top,$$

*where $k(\mathbf{q}, \mathbf{k}) = \exp\left(-\frac{\|\mathbf{q}-\mathbf{k}\|_1}{\lambda}\right)$ is the Laplacian kernel, and $\{\mathbf{k}_1, \dots, \mathbf{k}_N\} \subset \mathbb{R}^d$ is a fixed set of anchor points. Define the normalized feature mapping*

$$\mathbf{z}_i = \boldsymbol{\Sigma}^{-1/2}\left(\mathbf{g}_i - \frac{1}{N}\sum_{k=1}^{N} g_i^k \cdot \mathbf{1}\right) + \frac{1}{N}\mathbf{1},$$

*where $\boldsymbol{\Sigma}$ is the empirical covariance matrix over $\{\mathbf{g}_i\}$. Suppose the key set is sufficiently rich such that $\mathbf{q}_i \neq \mathbf{q}_j \Rightarrow \mathbf{g}_i \neq \mathbf{g}_j$. Assume further that for any $\mathbf{q}_i \neq \mathbf{q}_j$ and $c \in \mathbb{R}$, $\mathbf{g}_i - \mathbf{g}_j \neq c\mathbf{1}$ (no constant-vector degeneracy). Then the mapping $\mathbf{q}_i \mapsto \mathbf{z}_i \in \mathbb{R}^N$ is injective.*

*Proof.* We proceed by contradiction. Assume that $\mathbf{q}_i \neq \mathbf{q}_j$ but $\mathbf{z}_i = \mathbf{z}_j$.

For notational convenience, let $\mu_i = \frac{1}{N}\sum_{k=1}^{N} g_i^k$ denote the mean of the components of $\mathbf{g}_i$. We can then rewrite the normalized feature mapping as:

$$\mathbf{z}_i = \boldsymbol{\Sigma}^{-1/2}(\mathbf{g}_i - \mu_i \mathbf{1}) + \frac{1}{N}\mathbf{1}.$$

Expanding the assumed equality $\mathbf{z}_i = \mathbf{z}_j$, we obtain:

$$\boldsymbol{\Sigma}^{-1/2}(\mathbf{g}_i - \mu_i \mathbf{1}) + \frac{1}{N}\mathbf{1} = \boldsymbol{\Sigma}^{-1/2}(\mathbf{g}_j - \mu_j \mathbf{1}) + \frac{1}{N}\mathbf{1}.$$

Subtracting the constant vector $\frac{1}{N}\mathbf{1}$ from both sides yields:

$$\boldsymbol{\Sigma}^{-1/2}(\mathbf{g}_i - \mu_i \mathbf{1}) = \boldsymbol{\Sigma}^{-1/2}(\mathbf{g}_j - \mu_j \mathbf{1}).$$

Assuming non-degenerate cases where the covariance matrix $\boldsymbol{\Sigma}$ is strictly positive definite, $\boldsymbol{\Sigma}^{-1/2}$ is invertible. Left-multiplying both sides by $\boldsymbol{\Sigma}^{1/2}$ gives:

$$\mathbf{g}_i - \mu_i \mathbf{1} = \mathbf{g}_j - \mu_j \mathbf{1}.$$

Rearranging the terms gives:

$$\mathbf{g}_i - \mathbf{g}_j = (\mu_i - \mu_j)\mathbf{1}.$$

Let $c = \mu_i - \mu_j$, where $c \in \mathbb{R}$ is a scalar. Then the equation becomes:

$$\mathbf{g}_i - \mathbf{g}_j = c\mathbf{1}. \tag{10}$$

We now consider two mutually exclusive cases for the value of $c$:

**Case 1:** $c = 0$.
Equation (10) implies $\mathbf{g}_i = \mathbf{g}_j$. However, since we assumed $\mathbf{q}_i \neq \mathbf{q}_j$, this directly contradicts the proposition's assumption that $\mathbf{q}_i \neq \mathbf{q}_j \Rightarrow \mathbf{g}_i \neq \mathbf{g}_j$.

**Case 2:** $c \neq 0$.
Equation (10) implies $\mathbf{g}_i - \mathbf{g}_j = c\mathbf{1}$ for some scalar $c$. Since $\mathbf{q}_i \neq \mathbf{q}_j$, this directly contradicts the assumption of no constant-vector degeneracy.

In both cases, we arrive at a contradiction. Therefore, our initial assumption must be false. The only possibility is that $\mathbf{z}_i = \mathbf{z}_j \Rightarrow \mathbf{q}_i = \mathbf{q}_j$. Thus, the mapping is injective. $\qquad\square$

**Proposition 3** (Linear-Time Computation of Laplacian Feature Map via Nyström Approximation). *Let $\mathbf{q}_i \in \mathbb{R}^d$ be a query and $\{\tilde{\mathbf{k}}_1, \dots, \tilde{\mathbf{k}}_m\} \subset \mathbb{R}^d$ be a set of $m$ landmark keys sampled via the Nyström method, where $m \ll N$. Define the Laplacian kernel $k(\mathbf{q}, \mathbf{k}) = \exp\left(-\frac{\|\mathbf{q}-\mathbf{k}\|_1}{\lambda}\right)$, and let*

$$\tilde{\mathbf{g}}_i = \left[k(\mathbf{q}_i, \tilde{\mathbf{k}}_1), \dots, k(\mathbf{q}_i, \tilde{\mathbf{k}}_m)\right]^\top \in \mathbb{R}^m$$

*be the landmark kernel similarity vector for the $i$-th query. Assuming input dimension $d$ and the number of landmarks $m$ are constants with respect to the sequence length $N$, the normalized embedding*

$$\mathbf{z}_i = \mathbf{\Sigma}^{-1/2} \left( \tilde{\mathbf{g}}_i - \frac{1}{m} \sum_{j=1}^{m} \tilde{g}_i^j \cdot \mathbf{1} \right) + \frac{1}{m} \mathbf{1} \tag{11}$$

*can be computed in $\mathcal{O}(m)$ time and space. Consequently, computing the feature maps for all $N$ queries achieves an overall $\mathcal{O}(N)$ time and space complexity at inference time, provided that $\mathbf{\Sigma}^{-1/2}$ is approximated by an $m \times m$ diagonal whitening matrix estimated offline.*

*Proof.* The proof proceeds by analyzing the time and space complexity of computing the sequence of embeddings $\mathbf{z}_1, \ldots, \mathbf{z}_N$ at inference time, based on the $m$ sampled landmarks.

**Step 1: Computation of the landmark similarity vector $\tilde{\mathbf{g}}_i$.**
For each landmark key $\tilde{\mathbf{k}}_j$ ($j = 1, \ldots, m$), computing the Laplacian kernel element $\tilde{g}_i^j = \exp\left(-\frac{\|\mathbf{q}_i - \tilde{\mathbf{k}}_j\|_1}{\lambda}\right)$ requires:

1. Vector subtraction $\mathbf{q}_i - \tilde{\mathbf{k}}_j$ in $\mathbb{R}^d$, which takes $\mathcal{O}(d)$ time.

2. Computing the $L_1$ norm of the resulting difference vector, taking $\mathcal{O}(d)$ time.

3. Scalar division by $\lambda$ and exponentiation, taking $\mathcal{O}(1)$ time.

Since the input dimension $d$ is assumed to be constant, computing one element $\tilde{g}_i^j$ is bounded by $\mathcal{O}(d) = \mathcal{O}(1)$. Computing all $m$ elements for the vector $\tilde{\mathbf{g}}_i$ therefore takes $\mathcal{O}(m)$ time. Storing the resulting vector $\tilde{\mathbf{g}}_i$ requires $\mathcal{O}(m)$ space.

**Step 2: Centering the feature vector.**
Let $\mu_i = \frac{1}{m} \sum_{j=1}^{m} \tilde{g}_i^j$. Summing the $m$ elements of $\tilde{\mathbf{g}}_i$ and dividing by $m$ takes $\mathcal{O}(m)$ time. Subtracting this scalar mean from each component of $\tilde{\mathbf{g}}_i$ to compute the centered vector $\tilde{\mathbf{g}}_i - \mu_i \mathbf{1}$ takes an additional $\mathcal{O}(m)$ time. The space required for the centered vector is $\mathcal{O}(m)$.

**Step 3: Applying the whitening transformation.**
By assumption, the $m \times m$ matrix $\mathbf{\Sigma}^{-1/2}$ is approximated by a diagonal matrix computed offline. Let $\mathbf{D} = \mathrm{diag}(d_1, \ldots, d_m)$ be this diagonal approximation. The matrix-vector multiplication $\mathbf{D}(\tilde{\mathbf{g}}_i - \mu_i \mathbf{1})$ reduces to an element-wise multiplication (Hadamard product) between the diagonal entries of $\mathbf{D}$ and the centered vector. This element-wise operation takes $\mathcal{O}(m)$ time. Storing the diagonal of $\mathbf{D}$ requires $\mathcal{O}(m)$ space, and the resulting transformed vector also takes $\mathcal{O}(m)$ space. *(Note: Without the diagonal assumption, a full matrix-vector multiplication would require $\mathcal{O}(m^2)$ time).*

**Step 4: Adding the final bias term.**
Adding the scalar constant $\frac{1}{m}$ to each element of the resulting vector to obtain the final normalized embedding $\mathbf{z}_i \in \mathbb{R}^m$ requires $\mathcal{O}(m)$ time.

**Conclusion:**
For a single query $\mathbf{q}_i$, summing the time complexities yields $\mathcal{O}(m) + \mathcal{O}(m) + \mathcal{O}(m) + \mathcal{O}(m) = \mathcal{O}(m)$ inference time. Extending this operation to the entire sequence of $N$ queries results in a total inference time of $N \times \mathcal{O}(m) = \mathcal{O}(N \cdot m)$. Because the number of Nyström landmarks $m$ is a predefined parameter such that $m \ll N$, $m$ acts as a constant with respect to the sequence length. Thus, the overall computation of the feature maps scales linearly, achieving $\mathcal{O}(N)$ time and space complexity. $\qquad \square$

