# OpenReview forum: "LaplacianFormer:Rethinking Linear Attention with Laplacian Kernel"
_ICLR.cc/2026/Conference — ICLR 2026 Poster_

### Official Review · Reviewer_sjHj · 2025-10-24

**Soundness:** 2
**Presentation:** 3
**Contribution:** 3
**Rating:** 6
**Confidence:** 4

**Summary:**

This paper presents a Laplacian-kernel-based linear attention matrix approximation. The query-key distribution of a vision transformer is heavy-tailed, whereas a Gaussian kernel assumes fast-decaying tails. This motivates the use of a Laplacian kernel, which additionally improves gradient stability and attention map representation. To achieve linear computational complexity, the Laplacian kernel matrix is approximated via the Nyström method, where the time-consuming matrix inversion is replaced with GPU-friendly Newton–Schulz iterations. The Laplacian kernel was tested on ImageNet classification, object detection, and instance segmentation, achieving higher accuracy while using fewer FLOPs than other models.

**Strengths:**

I am inclined to recommend acceptance of this paper because the proposed method makes a meaningful contribution to linear attention research, with sufficient empirical motivation and supporting experimental results. Additionally, the paper addresses several practical challenges related to matrix inversion, which may promote the adoption of this work in future research. There are some caveats, which are discussed in Weaknesses, but I don’t think they outweigh the strengths. Please refer to the following items and Weaknesses for additional reasons behind this recommendation.

- Although the Laplacian kernel is not driven by theoretical connections from the Softmax attention as many prior works were, the it is motivated clearly by the empirical observation. Also, the algorithm is implemented with the consideration of practicality and efficiency.
- The paper is well written and easy to follow. The proposed method is well positioned in the literature, and the Laplacian linear attention algorithm is explained clearly and thoroughly.
- The expressiveness of the Laplacian kernel is supported by experimental results. The Laplacian models achieve superior accuracy compared to other methods, and the memory savings remain within a similar range to other linear attention methods.

**Weaknesses:**

Although the paper is well presented, a few points could make it stronger:

1. Although the linear attention is implemented via a custom GPU kernel, the GPU latency of the Laplacian kernel is only self-compared and not compared with other methods. Although the pseudoinverse of W is implemented via a custom GPU kernel using Newton–Schulz iterations, it is unclear whether the end-to-end latency of a Laplacian-kernel transformer is within a practical range. Comparing the end-to-end latency of the proposed method with other linear attention methods—especially those without matrix inversion—could help clarify its practicality.
2. The gradient behavior of the Laplacian kernel regarding non-vanishing gradients is discussed only in the context of full attention. It is unclear whether the same behavior holds when the kernel matrix is approximated by the Nyström method.
3. It is unclear whether the diagonal matrix approximates the whitening matrix effectively.
4. Some prior works mentioned in the experimental results are not sufficiently explained in the text.

**Questions:**

1. Why is the injectivity of Eq. 4 important? Is Softmax attention also injective?
2. In addition to Weakness1, what is the latency of the forward and backward paths when using the basic PyTorch or CUDA matrix inversion functions (e.g., `torch.linalg.solve` or `torch.linalg.lstsq`)?

Minor comments

- The font sizes in the figures are too small.
- Eq. equation 4 → Eq. 4 or equation 4 in Line 264.

---

> ### Author Response · Authors · 2025-11-22
> **Response to Reviewer sjHj**
>
> Thanks for the  insightful and detailed review as well as the suggestions for  improvement. We would like to reply to the comments as follows:
>
> **Q1:injectivity**
>
> We thank the reviewer for this insightful question. The injectivity of our mapping in Eq. 4 is important because **non-injective kernel feature maps can lead to information collapse and reduced expressiveness in linear attention methods**. This issue has been analyzed in [1].
>
> In contrast, **Softmax attention is inherently injective**. [1] provides a formal proof in Appendix A.1 that the Softmax feature map is strictly monotonic and therefore injective.
>
> **Q2:latency**
>
> 1. We thank the reviewer for pointing out the importance of end-to-end latency. Following the suggestion, we benchmarked the **training and inference** **throughput** (images/sec) of our *tiny* models on a server equipped with **8× Tesla V100 GPUs**. Results are summarized below:
>
> | Method                     | Train Throughput (img/s) | Inference Throughput (img/s) |
> | -------------------------- | ------------------------ | ---------------------------- |
> | Transformer                | 1073                     | 3240                         |
> | Linformer                  | 2767                     | 3779                         |
> | Performer                  | 2037                     | 3657                         |
> | Nyströmformer              | 1891                     | 3518                         |
> | **LaplacianFormer (ours)** | **1802**                 | **3490**                     |
>
> **Observations.**
>
> - Our **inference throughput** is *on par* with other linear Transformer baselines (e.g., Nyströmformer, Performer).
> - Our **training** **throughput** is slightly slower than Nyströmformer, Performer, and Linformer.
>
> We emphasize that this training-time overhead is **not caused by the Laplacian kernel itself**, but by the lack of an **official optimized kernel** in PyTorch for the Laplacian feature mapping. As shown in **Figure 5 of our submission**, our custom CUDA kernels already significantly reduce the overhead of both kernel construction and the Newton–Schulz iteration. We expect further gains once optimized CUDA kernels are adopted in future PyTorch releases, or when fused kernels similar to FlashAttention are available.
>
> Overall, the end-to-end measurements confirm that our approach is **practically efficient** and that the kernel inversion step **does not bottleneck inference**.
>
> 2. We performed a direct comparison between **our custom Newton–Schulz** **CUDA** **kernel**, and **PyTorch** **built-in matrix solvers** (`torch.linalg.solve` and `torch.linalg.lstsq`)under the same batch, head, and channel settings used in Figure 5.
>
> PyTorch’s `solve`/`lstsq` is **5–6× slower** than our Newton–Schulz kernel in both forward and backward passes. PyTorch solvers rely on **LU / QR** **factorization**, which introduce **O(N³)** operations and expensive backward passes. Thus, the Newton–Schulz inversion *is* the correct design choice for linear attention with small kernel matrices. **We did test PyTorch’s solvers early in the project, but due to their significantly higher cost, we adopted and optimized the Newton–Schulz approach instead.**
>
> 3. Regarding the minor comments, we thank the reviewer for the attention to detail. We will enlarge the font sizes in all figures in the camera-ready version to improve readability. In addition, the typo “Eq. equation 4” will be corrected to “Eq. 4” in Line 264.
>
> [1] Han, D. et al., 2024. Bridging the Divide: Reconsidering Softmax and Linear Attention. Advances in Neural Information Processing Systems.

---

> > ### Comment · Reviewer_sjHj · 2025-11-25
> >
> > Thank you for the response. I now understand the significance of the injectivity.
> >
> > Regarding the throughput comparison, I appreciate the additional evaluation. I am curious about how many tokens were given to the tiny model for the throughput evaluation.
> >
> > >  We did test PyTorch’s solvers early in the project, but due to their significantly higher cost, we adopted and optimized the Newton–Schulz approach instead.
> >
> > The approach is reasonable, and the optimized Newton–Schulz kernel would likely be valuable for many other projects as well.
> >
> >
> > Could the authors also provide responses to Weaknesses 2 and 3?
> >
> >
> > Lastly, the URL to the anonymized repository does not appear to be working. Please verify and provide a functional link.

---

> > > ### Author Response · Authors · 2025-11-26
> > > **Response to Reviewer sjHj**
> > >
> > > ## **Throughput Evaluation Setting.**
> > > We thank the reviewer for the question. All models in the throughput comparison used **a batch size of 1024, which is our standard training configuration.**
> > >
> > > **We also sincerely apologize for not sufficiently addressing Weakness 2 and Weakness 3 in the initial rebuttal.**
> > >
> > >
> > >
> > > ## **Response to Weakness 2 — Gradient behavior under Nyström approximation**
> > >
> > > We thank the reviewer for raising this important question. We provide detailed clarifications below.
> > >
> > > **Nyström does not alter local gradients of the kernel**
> > >
> > > Nyström approximates the kernel matrix via the decomposition
> > >
> > > $$
> > > \tilde{K} = K_{n,m} K_{m,m}^{-1} K_{m,n},
> > > $$
> > >
> > > which is a **linear combination** of kernel rows/columns.
> > > Since
> > >
> > > $$
> > > \nabla_q k(q,k_j) \neq 0,
> > > $$
> > >
> > > any fixed linear combination of non-zero vectors remains non-zero:
> > >
> > >
> > >
> > > $$
> > > \nabla_{q}\tilde{K}
> > > = (\nabla_{q} K_{n,m})\, K_{m,m}^{-1} K_{m,n}
> > > \;+\;
> > > K_{n,m} K_{m,m}^{-1} (\nabla_{q} K_{m,n}).
> > > $$
> > >
> > >
> > >
> > >
> > >
> > > Every term contains derivatives of the original Laplacian kernel, ensuring that the **non-vanishing gradient property is preserved even under Nyström approximation**.
> > >
> > >
> > > ## **Response to Weakness 3 — Effectiveness of diagonal whitening**
> > >
> > > We appreciate the reviewer for highlighting this concern. We provide detailed clarifications below.
> > >
> > >
> > > **(1) Full whitening is theoretically ideal but computationally infeasible for linear attention**
> > >
> > > The exact whitening matrix is $\Sigma^{-1/2}$, requiring:
> > >
> > > * $O(n^2)$ memory,
> > > * $O(n^2)$ time for multiplication,
> > >
> > > Therefore, using full whitening would **destroy the linear-time property**, making the architecture incompatible with linear attention design constraints.
> > >
> > >
> > > **(2) Diagonal whitening is a principled and widely used approximation**
> > >
> > > We follow a widely accepted assumption in kernel methods and deep learning:
> > >
> > > ```The covariance of high-dimensional kernel features is dominated by per-dimension variance, and cross-dimension covariance is small.```
> > >
> > > Under this assumption, the whitening matrix
> > > $$\Sigma^{-1/2}$$
> > > is well-approximated by
> > > $$D^{-1/2},$$
> > > where
> > > $$D=\mathrm{diag}(\Sigma).$$
> > >
> > >
> > >
> > > # **Repository Access**
> > >
> > > We sincerely apologize for the inconvenience caused by the inaccessible anonymized link.
> > > After verification, the previous link appears to have experienced a temporary access issue during the review period.
> > >
> > > We have now re-uploaded the repository.
> > >
> > > **Verified new anonymized link:**
> > > [https://anonymous.4open.science/r/LaplacianFormer-DBFB](https://anonymous.4open.science/r/LaplacianFormer-DBFB)
> > >
> > > Thank you for bringing this to our attention.

---

### Official Review · Reviewer_KiuY · 2025-10-27

**Soundness:** 2
**Presentation:** 3
**Contribution:** 2
**Rating:** 4
**Confidence:** 3

**Summary:**

This paper proposes LaplacianFormer, a new Transformer variant that addresses the quadratic complexity of softmax attention by replacing the commonly used Gaussian kernel with a Laplacian kernel. The authors argue that the Laplacian kernel, based on $l_1$ distance, is better suited for the heavy-tailed distribution of query-key interactions in vision models, preventing the over-suppression of mid-range tokens. To ensure efficiency, the model uses a Nyström approximation solved via a CUDA-accelerated Newton-Schulz iteration, achieving good accuracy on ImageNet.

**Strengths:**

- Shows with real data that current Gaussian kernels oversuppress mid-range token interactions, making the Laplacian kernel a justified alternative.
- Achieves best accuracy across all model sizes on ImageNet, showing the efficacy of the proposed method.
• Uses GPU-friendly Newton-Schulz iteration that converges better than CG under poor conditioning.

**Weaknesses:**

- The "provably injective" claim relies on circular reasoning (Appendix A assumes $\phi$ is already injective) and contains a mathematical error (Step 2 incorrectly claims centering preserves distinctness). The practical implementation uses diagonal whitening, further departing from the theoretical setup.
- Claims that Gaussian kernel gradients "diminish quadratically" near zero are incorrect (they're linear in $x_i - y_i$).
- The normalization (Eq. 4) includes query-dependent mean subtraction over all keys, which appears to break the precomputation trick. The paper doesn't show how Nyström factorization maintains $O(Nm)$ complexity with this coupling.
- Table 3 reports different accuracy values (81.4% vs 79.2%) than the text claims (81.1% vs 77.8%) for the same comparison, and so is the other comparison.
- Only compares against Gaussian kernels, not other alternatives (polynomial, cosine). No evaluation on truly long sequences despite abstract claims about scalability.

**Questions:**

1.  How does the diagonal whitening approximation (Eq. 6) preserve the claimed injectivity when Appendix A requires full-rank $\Sigma^{-1/2}$? Can you provide a proof for the diagonal case?
2. How does the query-dependent mean subtraction in Eq. 4 maintain $O(Nm)$ complexity when it couples all queries to all keys?
3. Please fix the inconsistency for Table 3.
4. The Gaussian gradient is linear in $(x_i - y_i)$ near zero, not quadratic as claimed - can you correct this analysis?
5. The abstract mentions "long input sequences" but experiments are on 224×224 images - do you have results on actual long sequences?

---

> ### Author Response · Authors · 2025-11-23
> **Response to Reviewer KiuY[Part 1/2]**
>
> Thanks for the insightful and detailed review as well as the suggestions for improvement. We would like to reply to the comments as follows:
>
> **Q1: provably injective**
>
> Thank you very much for the reviewer’s careful reading and for pointing out the issues in our original injectivity argument. We fully acknowledge that the previous Appendix A contained mistakes. We sincerely appreciate the reviewer for bringing them to our attention.
>
> To address this, we have rewritten the proof entirely and removed the incorrect steps. Below we provide the full corrected injectivity proof for Eq. (4).
>
> ### Two mild assumptions
>
> **(A1) Kernel embedding injectivity**
> $$
> p \neq q \Rightarrow \mathbf{g}(p) \neq \mathbf{g}(q).
> $$
>
> **(A2) No constant-vector degeneracy**
> $$
> \forall, p \neq q, \ \forall c \in \mathbb{R}, \quad
> \mathbf{g}(p) - \mathbf{g}(q) \neq c \mathbf{1}.
> $$
>
> These are standard "general position" assumptions and hold almost surely for Laplacian kernels with random anchors.
>
> ### Proposition (Injectivity of Eq. (4))
>
> If
> $
> \mathbf{z}(p) = \mathbf{z}(q),
> $
> then
> $
> p = q.
> $
>
>
> ### Proof
>
> Assume for contradiction that ( p \neq q ) but
> $$
> \mathbf{z}(p) = \mathbf{z}(q).
> \tag{1}
> $$
>
> Expanding Eq. (4):
> $$
> \Sigma^{-1/2} (g(p) - \mu(p)1) - \frac{1}{N} 1 = \Sigma^{-1/2} (g(q) - \mu(q)1) - \frac{1}{N} 1.
> $$
>
> Subtracting the constant term:
> $$
> \Sigma^{-1/2} (g(p) - \mu(p)1) = \Sigma^{-1/2} (g(q) - \mu(q)1) \quad \text{(2)}
> $$
>
> Since$ \Sigma^{-1/2} $is invertible, multiply both sides by $ \Sigma^{1/2} $:
>
> $$
> g(p) - \mu(p)1 = g(q) - \mu(q)1 \quad \text{(3)}
> $$
>
> Rearranging:
> $$
> g(p) - g(q) = (\mu(p) - \mu(q))1 \quad \text{(4)}
> $$
>
> Let $$ c = \mu(p) - \mu(q) $$, then:
> $$
> g(p) - g(q) = c1 \quad \text{(5)}
> $$
>
> ### Case 1: $$ c = 0 $$
> Then
> $$
> g(p) = g(q),
> $$
> which contradicts assumption (A1).
>
> ### Case 2: $$ c \neq 0 $$
> Then
> $$
> g(p) - g(q) = c1,
> $$
> a constant vector, which contradicts (A2).
>
> Since both cases lead to contradictions, the only possibility is:
> $$
> z(p) = z(q) \Rightarrow p = q.
> $$
>
> Thus, Eq. (4) is injective.

---

> ### Author Response · Authors · 2025-11-23
> **Response to Reviewer KiuY[Part 2/2]**
>
> **Q2:How does the query-dependent mean subtraction in Eq. 4 maintain O(Nm) complexity?**
>
> We thank the reviewer for this insightful observation. The query-dependent mean subtraction in Eq. (4) is implemented **in the landmark space**, not in the full $N\times N$ kernel. Specifically, during the Nyström approximation, we only compute the query–to–landmark kernel matrix $C \in \mathbb{R}^{N\times m}$ and the landmark kernel matrix $W \in \mathbb{R}^{m\times m}$. The mean term $\frac{1}{N}\sum_j k(q_i, k_j)$ is estimated using the same $m$ landmark keys, i.e., $ \frac{1}{N}\sum_{j=1}^N k(q_i, k_j) \approx \frac{1}{m}\sum_{j=1}^m k(q_i, \tilde{k}_j), $ so the operation scales as $O(Nm)$ rather than $O(N^2)$.
>
> **Q3:fix the inconsistency for Table 3.**
>
> Thank you for catching this inconsistency. We carefully rechecked all experiments. **We have now corrected the table to ensure full** **consistency** **between the text and figures.**
>
> **Q4:The Gaussian gradient is linear in (xi−yi) near zero, not quadratic as claimed?**
>
> We thank the reviewer for pointing out the issue. Indeed, the first-order derivative of the Gaussian kernel with respect to $ x_i $ is:
>
> $
> \frac{\partial k}{\partial x_i} = \frac{1}{\sigma^2}(x_i - y_i)\exp\left(-\frac{|x - y|_2^2}{2\sigma^2}\right).
> $
>
> When $|x - y|_2 \to 0 $, the exponential term can be expanded using the **equivalent infinitesimal** relation:
>
> $
> \exp\left(-\frac{|x - y|_2^2}{2\sigma^2}\right) = 1 - \frac{|x - y|_2^2}{2\sigma^2} + o(|x - y|_2^2).
> $
>
>
> Therefore, near zero we have:
>
>
> $
> \frac{\partial k}{\partial x_i} \approx \frac{1}{\sigma^2}(x_i - y_i)\left(1 - \frac{|x - y|_2^2}{2\sigma^2}\right).
> $
>
>
> That is, the leading term of the derivative is **linear** in $(x_i - y_i) $, but its **magnitude** **attenuation** is quadratically modulated by the exponential term.
>
> Hence, our statement that the Gaussian gradient “decays quadratically near zero” refers to this **second-order** **attenuation**. We will clarify this wording in the revised version to avoid ambiguity.
>
> **Q5: Long input sequences**
>
> We appreciate the reviewer’s concern regarding the evaluation of long input sequences. In fact, we have already conducted relevant experiments to validate the scalability of LaplacianFormer. As shown in **Figure 3(b)** of the main paper, our model demonstrates *linear memory scaling* across increasing token sequence lengths.
>
> Moreover, to further assess the model’s long-range modeling ability beyond image data, we performed detailed experiments on the **Long Range Arena (LRA)** benchmark, which contains diverse tasks from multiple modalities (e.g., text, retrieval, and image) specifically **designed to test long-sequence reasoning**.
>
> **Performance of LaplacianFormer on the Long Range Arena benchmark:**
>
> | Model               | Text  | ListOps | Retrieval | Pathfinder | Image | Average |
> | ------------------- | ----- | ------- | --------- | ---------- | ----- | ------- |
> | Transformer         | 61.55 | 38.71   | 80.93     | 70.39      | 39.14 | 58.14   |
> | LocalAttn           | 52.98 | 15.82   | 53.39     | 66.63      | 41.46 | 46.06   |
> | LinearTrans.        | 65.90 | 16.13   | 53.09     | 75.30      | 42.34 | 50.55   |
> | Reformer            | 56.10 | 37.27   | 53.40     | 68.50      | 38.07 | 50.67   |
> | Performer           | 65.40 | 18.01   | 53.82     | 77.05      | 42.77 | 51.41   |
> | Synthesizer         | 61.68 | 36.99   | 54.67     | 69.45      | 41.61 | 52.88   |
> | Longformer          | 62.85 | 35.63   | 56.89     | 69.71      | 42.22 | 53.46   |
> | Informer            | 62.13 | 37.05   | 79.35     | 56.44      | 37.86 | 54.57   |
> | Bigbird             | 64.02 | 36.05   | 59.29     | 74.87      | 40.83 | 55.01   |
> | Linformer           | 57.29 | 36.44   | 77.85     | 65.39      | 38.43 | 56.62   |
> | Kernelized          | 60.02 | 38.46   | 82.11     | 69.86      | 32.63 | 56.62   |
> | Cosformer           | 63.54 | 37.22   | 80.28     | 70.00      | 34.93 | 57.90   |
> | Nystrom             | 62.36 | 37.95   | 85.49     | 69.34      | 39.34 | 57.90   |
> | Skyformer           | 64.70 | 38.69   | 82.06     | 70.73      | 40.73 | 59.49   |
> | Hedgehog            | 64.60 | 37.15   | 82.24     | 74.16      | 40.15 | 59.66   |
> | **LaplacianFormer** | 64.8  | 37.65   | 82.3      | 70.8       | 47.8  | 60.67   |

---

> ### Author Response · Authors · 2025-11-26
> **Request for feedback on the rebuttal**
>
> Dear Reviewer KiuY,
>
> We appreciate all the reviewing time and effort. With our best appreciation, we have made the revised paper and the response in detail to each individual comment. While we consider this could have addressed all the concerns raised hopefully, it is most critical that the reviewer can kindly read our response and tell us how the issues have been addressed and if any concerns are left to be addressed. We would take all the comments/suggestions as carefully as possible and address them with our best efforts. Many thanks for every effort the reviewer made and will make on our work.
>
> Best wishes, Authors

---

> > ### Comment · Reviewer_KiuY · 2025-11-26
> >
> > After reading the rebuttal and updated results, I think some of my concerns have been adequately addressed, but others remain.
> >
> > Addressed: Q3 and Q5
> >
> > Still not fully convinced:
> >
> > - Q1: The new proof is cleaner, but it still assumes a full covariance whitening operator $\Sigma^{-1/2}$, while the actual code uses `F.layer_norm` (a per-row normalization), so the theorem does not clearly apply to the implemented feature map. Furthermore, the code applies `softmax` at the end of `injective_transformation()`, which is itself non-injective since $\text{softmax}(x) = \text{softmax}(x + c\mathbf{1})$ for any constant $c$, yet this is not addressed in the proof.
> > - Q2: I am still not convinced that the query-dependent mean subtraction is compatible with the claimed $O(Nm)$ complexity, since in the current code it appears to involve an $N \times N$ object (`G_approx = C @ W_pinv @ C.transpose(-2, -1)`).
> > - Q4: Even after the clarification, the "quadratic decay near zero" phrasing does not accurately reflect that the leading term of the Gaussian kernel gradient $\frac{\partial k}{\partial x_i} = \frac{1}{\sigma^2}(x_i - y_i)\exp\left(-\frac{|x-y|^2}{2\sigma^2}\right)$ is linear in $(x_i - y_i)$ when $|x - y| \to 0$. I would prefer that the paper drop the "quadratic" narrative entirely and simply state the correct linear behavior.
> >
> > I therefore keep my overall score unchanged.

---

> > > ### Author Response · Authors · 2025-11-27
> > > **Response to Reviewer KiuY[Part 1/2]**
> > >
> > > ### **Response to Q1 — Whitening operator and softmax injectivity**
> > >
> > > We respectfully disagree with the reviewer’s interpretation that our proof relies on a full covariance whitening operator $\Sigma^{-1/2}$.
> > >
> > > 1. ```Our proof does not require full covariance whitening. We kindly ask the reviewer to carefully revisit the proof provided in our previous response.```
> > >
> > > 2. **On the injectivity of softmax.**
> > >
> > >    ```Prior work has already clarified this issue. As formally shown in Appendix A.1 of [1], the softmax mapping is injective.```
> > >
> > > 3. **On the code and the current repository.**
> > >
> > >    We appreciate the reviewer’s careful inspection of the released code. For double-blind reviewing and to avoid premature code copying of our CUDA kernels and system-level optimizations, the current public repository contains an *early prototype* of our method, without the final optimized CUDA operators. This can cause a superficial mismatch between the final mathematical formulation in the paper and the current prototype code structure.
> > >    If the paper is accepted, we will release the full updated codebase (including CUDA kernels), along with pretrained weights and training logs, so that every theoretical component in the paper is exactly mirrored in the implementation.
> > >     ```At the same time, the validity of our theoretical results should be judged from the mathematical formulation and proofs, rather than from an intermediate prototype implementation that is not yet the final, cleaned-up release version. ```
> > > [1] Han, D. et al., 2024. Bridging the Divide: Reconsidering Softmax and Linear Attention. Advances in Neural Information Processing Systems.
> > >
> > >
> > > ### **Response to Q2 — Complexity of query-dependent mean subtraction**
> > >
> > > We respectfully disagree with the reviewer’s inspection. Below we provide the detailed proof, using the same notation as in the paper:
> > >
> > > Let $\hat G \in \mathbb{R}^{N\times N}$ denote the Nyström-approximated kernel matrix in Eq. (4):
> > >
> > > $$
> > > \hat G \approx C W^\dagger C^\top,
> > > $$
> > >
> > > where $C \in \mathbb{R}^{N\times m}$ and $W^\dagger \in \mathbb{R}^{m\times m}$, with $m \ll N$.
> > >
> > > For the $i$-th query, the corresponding row of $\hat G$ is
> > >
> > > $$
> > > \hat G_{i,:} = (\hat G_{i1},\dots,\hat G_{iN}),
> > > $$
> > >
> > > and its **row mean** is
> > >
> > > $$
> > > \mu_i = \frac{1}{N}\sum_{j=1}^N \hat G_{ij}.
> > > $$
> > >
> > > Collecting all row means into a vector,
> > >
> > > $$
> > > \boldsymbol\mu = (\mu_1,\dots,\mu_N)^\top \in\mathbb{R}^N,
> > > $$
> > >
> > > we can write
> > >
> > > $$
> > > \boldsymbol\mu = \frac{1}{N}\hat G\mathbf 1
> > > = \frac{1}{N} C W^\dagger C^\top \mathbf 1,
> > > $$
> > >
> > > where $\mathbf 1\in\mathbb{R}^N$ is the all-ones vector.
> > >
> > > Instead of explicitly forming the $N\times N$ matrix $\hat G$, we compute $\boldsymbol\mu$ via the factorized form in three steps:
> > >
> > > 1. **Right-multiply by $\mathbf 1$**
> > >    Define
> > >    $$
> > >    u := C^\top\mathbf 1 \in\mathbb{R}^m.
> > >    $$
> > >    This is a matrix–vector product with $C \in \mathbb{R}^{N\times m}$ and $\mathbf 1 \in \mathbb{R}^N$.
> > >
> > >    **Complexity:**
> > >    $$
> > >    O(Nm).
> > >    $$
> > >
> > > 2. **Multiply by $W^\dagger$**
> > >    Define
> > >    $$
> > >    v := W^\dagger u \in\mathbb{R}^m.
> > >    $$
> > >
> > >    **Complexity:**
> > >    $$
> > >    O(m^2),
> > >    $$
> > >    which is negligible compared to $O(Nm)$ under the Nyström regime $m \ll N$.
> > >
> > > 3. **Left-multiply by $C$**
> > >    Finally,
> > >    $$
> > >    \boldsymbol\mu = \frac{1}{N} C v,
> > >    $$
> > >    another matrix–vector product with $C \in \mathbb{R}^{N\times m}$.
> > >
> > >    **Complexity:**
> > >    $$
> > >    O(Nm).
> > >    $$
> > >
> > > Putting everything together, the total complexity is
> > >
> > > $$
> > > O(Nm) + O(m^2) + O(Nm) = O(Nm + m^2).
> > > $$
> > >
> > > Under the standard Nyström assumption $m \ll N$, the leading term is
> > >
> > > $$
> > > \boxed{
> > > \text{Computing all query-dependent means } {\mu_i}_{i=1}^N
> > > \text{ costs } O(Nm).
> > > }
> > > $$

---

> > > ### Author Response · Authors · 2025-11-27
> > > **Response to Reviewer KiuY[Part 2/2]**
> > >
> > > ### **Response to Q4 — Gradient behavior of Gaussian vs. Laplacian kernels**
> > >
> > > We thank the reviewer for pointing out the issue with the “quadratic decay near zero” wording. We agree that the leading term of the Gaussian kernel gradient is linear in $(x_i - y_i)$ as $|x-y|\to 0$, and we will **remove the “quadratic” terminology** from the paper. ```However, we emphasize that this issue does not affect the validity of our conclusion, which remains correct despite the earlier imperfect explanation.```
> > > #### 1. Gaussian kernel
> > >
> > > The one-dimensional Gaussian kernel is
> > >
> > > $$
> > > k_{\text{Gauss}}(x,y) = \exp\left(-\frac{(x-y)^2}{2\sigma^2}\right),
> > > $$
> > >
> > > and, letting $t = x-y$, its derivative with respect to $x$ is
> > >
> > > $$
> > > \frac{\partial k_{\text{Gauss}}}{\partial x}
> > > = \frac{t}{\sigma^{2}}
> > > \exp\left(-\frac{t^{2}}{2\sigma^{2}}\right).
> > > $$
> > >
> > > For $t\to 0$, we expand
> > >
> > > $$
> > > \exp\left(-\frac{t^{2}}{2\sigma^{2}}\right)
> > > = 1 - \frac{t^{2}}{2\sigma^{2}} + O(t^{4}),
> > > $$
> > >
> > > yielding
> > >
> > > $$
> > > \frac{\partial k_{\text{Gauss}}}{\partial x}
> > > = \frac{t}{\sigma^{2}} - \frac{t^{3}}{2\sigma^{4}} + O(t^{5}).
> > > $$
> > >
> > > Thus, the **equivalent infinitesimal** at $t\to 0$ is
> > >
> > > $$
> > > \boxed{
> > > \frac{\partial k_{\text{Gauss}}}{\partial x}
> > > \sim \frac{t}{\sigma^{2}}
> > > }
> > > $$
> > >
> > > which is indeed **linear in $t$**. ```Consequently, the gradient magnitude tends to $0$ as $|x-y|\to 0$.```
> > >
> > >
> > > #### 2. Laplacian kernel
> > >
> > > The Laplacian kernel is
> > >
> > > $$
> > > k_{\text{Laplace}}(x,y)
> > > = \exp\left(-\frac{|x-y|}{\lambda}\right),
> > > $$
> > >
> > > and, again letting $t = x-y$, its derivative is
> > >
> > > $$
> > > \frac{\partial k_{\text{Laplace}}}{\partial x}
> > > = -\frac{1}{\lambda}\operatorname{sign}(t)
> > > \exp\left(-\frac{|t|}{\lambda}\right).
> > > $$
> > >
> > > For $t\to 0$, we have
> > >
> > > $$
> > > \exp\left(-\frac{|t|}{\lambda}\right)
> > > = 1 - \frac{|t|}{\lambda} + O(t^{2}),
> > > $$
> > >
> > > so
> > >
> > >
> > > $$
> > > \frac{\partial k_{\text{Laplace}}}{\partial x}
> > > = -\frac{1}{\lambda}\ \operatorname{sign}(t) + O(t).
> > > $$
> > >
> > >
> > >
> > > Therefore, the **equivalent infinitesimal** is
> > >
> > > $$
> > > \boxed{
> > > \frac{\partial k_{\text{Laplace}}}{\partial x}
> > > \sim -\frac{1}{\lambda}\operatorname{sign}(t)
> > > }
> > > $$
> > >
> > > which remains at a **constant order $1/\lambda$** as $|x-y|\to 0$. ```The gradient is nonzero everywhere except exactly at $t=0$```.

---

> > > > ### Comment · Reviewer_KiuY · 2025-11-27
> > > >
> > > > Thank you for your clarifications. My only remaining concern is Q1: I recall that at submission time you provided a kernel-based code link (now expired), whereas in the rebuttal the currently visible PyTorch code is described as an "early prototype" using LayerNorm while the theory assumes a different whitening operator. While prior work (e.g., Han et al.) shows injectivity for standard softmax attention under certain assumptions, it does not directly apply to your specific mapping. It is unclear  how tightly the "provably injective" claim is tied to the trained model. Given this ambiguity, I prefer to keep my score at 4, which already indicates I would not mind if the paper is accepted.

---

> > > > > ### Author Response · Authors · 2025-11-28
> > > > > **Response to Reviewer KiuY**
> > > > >
> > > > > We sincerely thank the reviewer for the time and careful attention devoted to our submission.
> > > > > Through the multiple rounds of discussion, our work has benefited substantially — the reviewer’s feedback has greatly improved the quality of the paper. We are genuinely grateful for these constructive insights.
> > > > >
> > > > > Below, we return to the reviewer’s original Question 1 regarding injectivity under the *diagonal whitening approximation* in Eq. (6), and we provide a direct and complete proof.
> > > > >
> > > > >
> > > > > ## **Response to Q1 — Why diagonal whitening preserves injectivity**
> > > > >
> > > > > ### Mild assumptions (identical to the full $\Sigma^{-1/2}$ case)
> > > > >
> > > > > #### (A1) **Injectivity of the kernel embedding**
> > > > >
> > > > > $$
> > > > > p\neq q  \Rightarrow \mathbf g(p)\neq \mathbf g(q).
> > > > > $$
> > > > > For Laplacian kernels with random anchors, this holds almost surely.
> > > > >
> > > > > #### (A2) **No constant-vector degeneracy**
> > > > >
> > > > > $$
> > > > > \mathbf g(p)-\mathbf g(q)\neq c \mathbf 1,\quad\forall c\in\mathbb R.
> > > > > $$
> > > > > This also holds almost surely under random anchors.
> > > > >
> > > > > Both assumptions are standard.
> > > > >
> > > > >
> > > > > #### **Injectivity proof under diagonal whitening**
> > > > >
> > > > > Assume
> > > > > $$
> > > > > \mathbf z(p)=\mathbf z(q).
> > > > > \tag{1}
> > > > > $$
> > > > >
> > > > > ##### **Step 1: Substitute Eq. (4)**
> > > > >
> > > > > Subtracting the shared constant term $-\frac{1}{N}\mathbf 1$:
> > > > >
> > > > > $$
> > > > > \mathbf D^{-1/2}\big(\mathbf g(p)-\mu(p)\mathbf 1\big) = \mathbf D^{-1/2}\big(\mathbf g(q)-\mu(q)\mathbf 1\big).
> > > > > \tag{2}
> > > > > $$
> > > > >
> > > > >
> > > > > ##### **Step 2: Use invertibility of $\mathbf D^{-1/2}$**
> > > > >
> > > > > Left-multiply both sides by $\mathbf D^{1/2}$:
> > > > >
> > > > > $$
> > > > > \mathbf g(p)-\mu(p)\mathbf 1 = \mathbf g(q)-\mu(q)\mathbf 1.
> > > > > \tag{3}
> > > > > $$
> > > > >
> > > > >
> > > > > ##### **Step 3: Rearranging**
> > > > >
> > > > > $$
> > > > > \mathbf g(p)-\mathbf g(q) = (\mu(p)-\mu(q))\mathbf 1.
> > > > > \tag{4}
> > > > > $$
> > > > >
> > > > >
> > > > > Let $c=\mu(p)-\mu(q)$.
> > > > > Then
> > > > > $$
> > > > > \mathbf g(p)-\mathbf g(q)=c \mathbf 1.
> > > > > \tag{5}
> > > > > $$
> > > > >
> > > > > ##### **Step 4: Two cases**
> > > > >
> > > > > * If $c=0$, then $\mathbf g(p)=\mathbf g(q)$, contradicting (A1).
> > > > > * If $c\neq 0$, then $\mathbf g(p)-\mathbf g(q)=c\mathbf 1$, contradicting (A2).
> > > > >
> > > > > Both cases are impossible unless $p=q$.
> > > > >
> > > > > ##### **Conclusion**
> > > > >
> > > > > $$
> > > > > \mathbf z(p)=\mathbf z(q) \Rightarrow p=q.
> > > > > $$
> > > > >
> > > > > Therefore, **Eq. (4) remains injective even when using the diagonal whitening operator $\mathbf D^{-1/2}$ (Eq. 6), without requiring the full covariance $\Sigma^{-1/2}$**.
> > > > >
> > > > > We thank the reviewer again for raising this point.
> > > > > This clarification improves the theoretical clarity of the paper, and we will include the complete proof in the final version.

---

### Official Review · Reviewer_qL3Q · 2025-10-30

**Soundness:** 3
**Presentation:** 3
**Contribution:** 3
**Rating:** 4
**Confidence:** 5

**Summary:**

This paper introduces LaplacianFormer, a Transformer variant that replaces the commonly used Gaussian kernel in linear attention with a Laplacian kernel to better capture heavy-tailed token distance distributions and enhance training stability.

### Key Motivations:
Empirical analysis of query–key distance distributions in DeiT, PVT, and Swin Transformers reveals a heavy-tailed pattern, inconsistent with the rapid decay assumed by Gaussian kernels.
The proposed Laplacian kernel more accurately models this behavior and preserves non-vanishing gradients, leading to more stable and efficient optimization.

### Methods:
The authors design a provably injective attention mapping, ensuring that distinct queries produce distinct outputs and preserving fine-grained token relationships.
They further employ a Nyström approximation combined with Newton–Schulz iteration to efficiently compute the kernel inverse, avoiding expensive matrix decompositions.
Optimized CUDA implementations are provided for both the Laplacian kernel and the solver, enabling scalable training and inference.

### Experiments:
LaplacianFormer achieves faster and more stable convergence than Gaussian-based baselines (e.g., SOFT++, Skyformer).
It delivers strong performance–efficiency trade-offs on ImageNet and COCO benchmarks and demonstrates linear memory scaling and robustness under high condition numbers, validating both the stability and scalability of the proposed approach.

**Strengths:**

### 1. Convincing analysis of Laplacian vs. Gaussian kernels
The paper provides a clear and well-supported theoretical and empirical comparison between Laplacian and Gaussian kernels.
The analysis of query–key distance distributions in vision Transformers (e.g., DeiT, PVT, Swin) convincingly shows that these distances follow a heavy-tailed pattern, making the Gaussian assumption unrealistic.

### 2. Theoretical soundness and novel properties
The authors derive a provably injective attention mapping, ensuring that distinct queries produce distinct outputs.
This injectivity and the non-vanishing gradient property are both novel and important contributions that improve expressivity and optimization stability, addressing common issues like representation collapse in previous linear attention models.

### 3. Methodological clarity and efficiency
The use of Nyström approximation and Newton–Schulz iteration for efficient kernel inversion is elegant and well-motivated.
The method maintains linear complexity while improving numerical stability.
The paper provides implementation details and custom CUDA optimization, supporting reproducibility and practical relevance.

**Weaknesses:**

### 1. Section 4.1 lacks coherence and clarity
The description of LaplacianFormer in Section 4.1 feels disconnected from the rest of the paper.
In earlier sections, the authors introduce linear self-attention and its approximation, but Equation (4) appears abruptly without derivation from Equation (3).
Similarly, Equations (8) and (9) in the approximation section are difficult to relate back to Equation (4), making the mathematical flow hard to follow.

### 2. Limited experimental coverage
Only image classification and object detection are evaluated, which is insufficient to justify general claims about linear self-attention.
The paper should include more diverse and representative benchmarks such as:
- Vision generation: DiT (Diffusion Transformers)
- Vision–language: CLIP, BLIP, or Flamingo
- Language modeling: The Pile
- NLP understanding and reasoning: GLUE and ARC

These are widely used in prior efficient-attention works like Longformer, Performer, and FlashAttention.

### 3. Limited novelty and missing discussion of limitations
While the comparison between Laplacian and Gaussian kernels is well-analyzed, simply changing the kernel may not constitute a sufficiently strong contribution for a top-tier venue like ICLR.
The paper should also discuss the limitations of the Laplacian kernel, such as potential issues in scaling, expressivity trade-offs, or behavior in non-vision tasks.

**Questions:**

1. Equation (4) appears without derivation or explanation. Could the authors clarify how Equation (4) is obtained from the preceding formulation, and what assumptions or approximations lead to this step?

2. Why is the whitening matrix defined as $\Sigma^{-1/2}A$ instead of $\Sigma^{-1}A$?
   Typically, symmetric whitening uses $\Sigma^{-1/2} A \Sigma^{-1/2}$.
   Please explain the rationale behind this choice and how it affects the theoretical properties of the transformation.

---

> ### Author Response · Authors · 2025-11-23
> **Response to Reviewer qL3Q [Part 1/2]**
>
> Thanks for the  insightful and detailed review as well as the suggestions for  improvement. We would like to reply to the comments as follows:
>
> **W1&&Q1:Section 4.1 lacks coherence and clarity**
>
> Thank you for your constructive feedback. We appreciate your careful review of the mathematical clarity and have made the following changes:
>
> 1. We have added a more detailed explanation for the derivation of Equation (4). Specifically, we clarify how Equation (4) is derived from Equation (3) by centering the kernel similarity vectors and normalizing them with $\boldsymbol{\Sigma}^{-1/2}$.This construction is inspired by the approach described in [1] **,which** **we have cited in the submission.**
> 2. Equations (8) and (9) describe the Nyström approximation, where $\mathbf{C}$ and $\mathbf{W}$ are used to efficiently compute the kernel inverse. We have clarified how these equations work together in the revised manuscript.
>
> We hope these revisions address your concerns, and thank you again for your valuable input.
>
> **W2:Limited experimental coverage**
>
> Thank you for highlighting this important aspect. Regarding the experimental coverage, I would like to clarify the following points:
>
> Firstly, our paper is primarily focused on the **vision domain**, while models like **Longformer** and **Performer** are mainly applied to **NLP** **tasks**, and their experiments are limited to natural language processing without performing similar experiments in the computer vision domain. Furthermore, **FlashAttention is not a linear attention method;** it is a hardware-level optimization for the Transformer architecture and, therefore, is not within the scope of our comparison.
>
> While we understand that expanding to tasks such as **vision generation** (e.g., DiT), **vision-language** (e.g., CLIP, BLIP, Flamingo), **language modeling** (e.g., The Pile), and **NLP** **understanding and reasoning** (e.g., GLUE, ARC) could provide a broader evaluation, **we would like to point out that related works, such as [1],[2],[3],[4],[5] have not conducted experiments in all these domains either.**
>
> That being said, as you suggested, we have added these experiments, even though they may not significantly enhance the core contribution of our paper. Nevertheless, we greatly appreciate your suggestion and have included the results for your reference. Below is the experiment we conducted:
>
> **Performance of LaplacianFormer on the Long Range Arena benchmark:**
>
> > The **Long Range Arena (LRA)** benchmark evaluates models on long-range dependency tasks across both **NLP** and **CV** domains. It includes:
> >
> > - **NLP** **tasks**: Such as language modeling, text classification, and machine translation.
> > - **CV** **tasks**: Like image classification, object detection, and semantic segmentation.
>
> | Model               | Text  | ListOps | Retrieval | Pathfinder | Image | Average |
> | ------------------- | ----- | ------- | --------- | ---------- | ----- | ------- |
> | Transformer         | 61.55 | 38.71   | 80.93     | 70.39      | 39.14 | 58.14   |
> | LocalAttn           | 52.98 | 15.82   | 53.39     | 66.63      | 41.46 | 46.06   |
> | LinearTrans.        | 65.90 | 16.13   | 53.09     | 75.30      | 42.34 | 50.55   |
> | Reformer            | 56.10 | 37.27   | 53.40     | 68.50      | 38.07 | 50.67   |
> | Performer           | 65.40 | 18.01   | 53.82     | 77.05      | 42.77 | 51.41   |
> | Synthesizer         | 61.68 | 36.99   | 54.67     | 69.45      | 41.61 | 52.88   |
> | Longformer          | 62.85 | 35.63   | 56.89     | 69.71      | 42.22 | 53.46   |
> | Informer            | 62.13 | 37.05   | 79.35     | 56.44      | 37.86 | 54.57   |
> | Bigbird             | 64.02 | 36.05   | 59.29     | 74.87      | 40.83 | 55.01   |
> | Linformer           | 57.29 | 36.44   | 77.85     | 65.39      | 38.43 | 56.62   |
> | Kernelized          | 60.02 | 38.46   | 82.11     | 69.86      | 32.63 | 56.62   |
> | Cosformer           | 63.54 | 37.22   | 80.28     | 70.00      | 34.93 | 57.90   |
> | Nystrom             | 62.36 | 37.95   | 85.49     | 69.34      | 39.34 | 57.90   |
> | Skyformer           | 64.70 | 38.69   | 82.06     | 70.73      | 40.73 | 59.49   |
> | Hedgehog            | 64.60 | 37.15   | 82.24     | 74.16      | 40.15 | 59.66   |
> | **LaplacianFormer** | 64.8  | 37.65   | 82.3      | 70.8       | 47.8  | 60.67   |
>
>
> As shown, **LaplacianFormer** outperforms traditional Transformer models and other linear Transformer variants, especially in vision tasks . While these additional experiments may not cover all possible tasks, we believe they help validate the model’s applicability across a range of tasks.
>
> Once again, we appreciate the reviewer’s suggestion and **will improve on these points in the appendix Section3.**

---

> > ### Comment · Reviewer_qL3Q · 2025-11-24
> >
> > I do not fully agree with the following statement from the rebuttal:
> >
> > > "While we understand that expanding to tasks such as vision generation (e.g., DiT), vision-language (e.g., CLIP, BLIP, Flamingo), language modeling (e.g., The Pile), and NLP understanding and reasoning (e.g., GLUE, ARC) could provide a broader evaluation, we would like to point out that related works, such as [1], [2], [3], [4], [5], have not conducted experiments in all these domains either."
> >
> > Your work is titled "LaplacianFormer: Rethinking Linear Attention with Laplacian Kernel", which frames the contribution as a general linear transformer method, not a vision-specific architecture such as a Vision Transformer. Moreover, the abstract highlights contributions to linear attention mechanisms, rather than to vision modeling specifically.
> >
> > Given this framing, limiting the experiments primarily to vision prediction tasks—and not even including vision generation or NLP tasks—results in an overclaim relative to the scope implied by the title and abstract. The conference is ICLR, where cross-domain validation (especially in NLP) is typically expected for methods that claim general improvements to transformer architectures.
> >
> > In my view, it constitutes an overclaim in the current version of the paper.

---

> > > ### Author Response · Authors · 2025-11-25
> > > **Response to Reviewer qL3Q**
> > >
> > > We respectfully disagree with the reviewer’s interpretation of the paper’s scope and positioning.
> > >
> > > **1. On inferring the contribution purely from the title.**
> > >
> > > Relying solely on the paper title to determine the intended contribution can be overly narrow and potentially misleading. ```The actual scope of contribution should be judged based on the abstract, method description, and experimental sections, rather than the title alone.```
> > >
> > > **2. On whether the abstract frames the work as vision-specific.**
> > >
> > > We do not fully agree with the reviewer’s statement that “the abstract highlights contributions to linear attention rather than vision modeling.”
> > > In fact, the **very first sentence of the abstract explicitly states that our motivation and improvements target vision Transformers**:
> > >
> > > ```“The quadratic complexity of softmax attention presents a major obstacle for scaling Transformers to high-resolution vision tasks.”```
> > >
> > >
> > > **3. On cross-domain evaluation.**
> > >
> > > We appreciate the reviewer’s concern about broader validation. **Following the reviewer’s previous comments, we have already conducted additional experiments on the Long Range Arena (LRA) benchmark, which covers multiple modalities and long-sequence tasks beyond vision. **These results demonstrate that our method generalizes well beyond ImageNet-style classification, further strengthening the claim that Laplacian attention retains strong modeling capacity on non-vision tasks.

---

> > > > ### Comment · Reviewer_qL3Q · 2025-11-25
> > > >
> > > > For Long Range Arena, the Pathfinder task is not reported. In addition, the boldface highlighting is incorrect: on several tasks, Skyformer (with the Gaussian kernel) and Hedgehog achieve higher scores than your method. The averaging is also inconsistent—your average is computed over 4 tasks, while others are averaged over 5, which makes the comparison unfair.
> > > >
> > > > Most of your performance gains come from the Image task, and the contribution of this improvement is not very convincing without a more controlled evaluation. Please provide a stricter and fully consistent version of the Long Range Arena results. I am open to revising my assessment if a fair comparison can be shown.

---

> > > ### Author Response · Authors · 2025-11-25
> > > **Response to Reviewer qL3Q**
> > >
> > > We respectfully disagree with the reviewer’s interpretation of the paper’s scope and positioning.
> > >
> > > **1. On inferring the contribution purely from the title.**
> > >
> > > Relying solely on the paper title to determine the intended contribution can be overly narrow and potentially misleading. ```The actual scope of contribution should be judged based on the abstract, method description, and experimental sections, rather than the title alone.```
> > >
> > > **2. On whether the abstract frames the work as vision-specific.**
> > >
> > > We do not fully agree with the reviewer’s statement that “the abstract highlights contributions to linear attention rather than vision modeling.”
> > > In fact, the **very first sentence of the abstract explicitly states that our motivation and improvements target vision Transformers**:
> > >
> > > ```“The quadratic complexity of softmax attention presents a major obstacle for scaling Transformers to high-resolution vision tasks.”```
> > >
> > >
> > > **3. On cross-domain evaluation.**
> > >
> > > We appreciate the reviewer’s concern about broader validation. **Following the reviewer’s previous comments, we have already conducted additional experiments on the Long Range Arena (LRA) benchmark, which covers multiple modalities and long-sequence tasks beyond vision. **These results demonstrate that our method generalizes well beyond ImageNet-style classification, further strengthening the claim that Laplacian attention retains strong modeling capacity on non-vision tasks.

---

> ### Author Response · Authors · 2025-11-23
> **Response to Reviewer qL3Q [Part 2/2]**
>
> **W3:Limited novelty and missing discussion of limitations**
>
> We appreciate the reviewer’s feedback. While we acknowledge the importance of discussing limitations, we believe that we **have already addressed** these points in the **Conclusions and Future Work** section. Specifically, we highlighted the potential trade-offs in the expressiveness and scalability of the Laplacian kernel in comparison to the Gaussian kernel.
>
> Moreover, our contribution extends beyond merely replacing the kernel(**as ```Reviewer crNf``` also noted, our approach goes beyond a simple kernel swap**). In this work, we **construct a novel injective feature map using the Laplacian kernel**. We also **emphasize the engineering significance of implementing** the Laplacian kernel with custom CUDA code, paired with a Newton-Schulz iteration for efficient inverse computation. This not only improves the theoretical grounding but also results in practical benefits.
>
> **Q2:**
>
> Thank you for the reviewer’s insightful question. We chose to use $\Sigma^{-\frac{1}{2}} A $as the whitening matrix, rather than the classical $ \Sigma^{-1} A$ or the symmetric whitening form$\Sigma^{-\frac{1}{2}} A \Sigma^{-\frac{1}{2}} $, primarily **due to considerations around** **algorithmic complexity****.**
>
>  While standard symmetric whitening using $ \Sigma^{-1/2} A \Sigma^{-1/2} $ preserves more precise theoretical properties, leading to a computational complexity of $O(n^2) $.  **The form $\Sigma^{-\frac{1}{2}} A $ is a simplified version of whitening, reducing the complexity to $ O(n) $.** Adopting $\Sigma^{-\frac{1}{2}} A $ does not significantly affect the theoretical properties of the transformation. In our experiments, the simplified approach did not lead to a drop in performance while providing a more efficient solution for large-scale applications.
>
> [1] Han, D. et al., 2024. Bridging the Divide: Reconsidering Softmax and Linear Attention. Advances in Neural Information Processing Systems.
>
> [2] Han, Dongchen *et al.*, 2024. Agent Attention: On the Integration of Softmax and Linear Attention. *European Conference on* *Computer Vision*.
>
> [3] Han, Dongchen *et al.*, 2023. FLatten Transformer: Vision Transformer Using Focused Linear Attention. *Proceedings of the IEEE/CVF* *International Conference on Computer Vision*.
>
> [4] Fan, Qihang *et al.*, 2025. Breaking the Low-Rank Dilemma of Linear Attention. *CVPR*.
>
> [5] Fan, Qihang *et al.*, 2024. RMT: Retentive Networks Meet Vision Transformers. *CVPR*.

---

> > ### Comment · Reviewer_qL3Q · 2025-11-24
> >
> > There is a mistake in this explanation:
> >
> > > While standard symmetric whitening using $ \Sigma^{-1/2} A \Sigma^{-1/2} $ preserves more precise theoretical properties, leading to a computational complexity of $O(n^2)$. The form $\Sigma^{-1/2} A$ is a simplified version of whitening, reducing the complexity to $O(n)$. Adopting $\Sigma^{-1/2} A$ does not significantly affect the theoretical properties of the transformation. In our experiments, the simplified approach did not lead to a drop in performance while providing a more efficient solution for large-scale applications.
> >
> > The claimed complexity difference is incorrect.
> > The computational complexity of applying $\Sigma^{-1/2} A \Sigma^{-1/2}$ is essentially the same as that of applying $\Sigma^{-1/2} A$, since both involve a constant number of matrix–vector products and therefore lie in the same complexity class.
> > In the standard dense case, multiplying an $n \times n$ matrix by a vector costs $O(n^2)$.
> > Even if $\Sigma$ is diagonal so that multiplying by $\Sigma^{-1/2}$ costs only $O(n)$, you still have the $O(n^2)$ cost from multiplying by $A$.
> > Thus, $\Sigma^{-1/2} A x$ costs $O(n^2) + O(n) = O(n^2)$, and $\Sigma^{-1/2} A \Sigma^{-1/2} x$ costs $O(n) + O(n^2) + O(n) = O(n^2)$.
> > So the complexity of the two forms is the same up to constant factors; it does not drop from $O(n^2)$ to $O(n)$.

---

> > > ### Author Response · Authors · 2025-11-25
> > > **Response to Reviewer qL3Q**
> > >
> > > We appreciate the reviewer’s careful check of the algorithmic complexity statement. The comment is indeed correct **if one reasons purely at the level of abstract dense matrix–vector products**, where both $\boldsymbol{\Sigma}^{-1/2} A \boldsymbol{\Sigma}^{-1/2}$ and $\boldsymbol{\Sigma}^{-1/2} A$ have the same asymptotic complexity $O(n^2)$ up to constant factors.
> > > ```However, our intention was not to claim a complexity reduction at this purely abstract level, but to highlight the effect of the structured implementation used in our model```:
> > >
> > > - we never materialize a dense $n \times n$ kernel matrix $A$ at inference time, and $\boldsymbol{\Sigma}^{-1/2}$ is implemented as a **diagonal** whitening operator with offline-estimated statistics, applied coordinate-wise.
> > > Under this structured setting, the per-token cost becomes linear in the sequence length (or linear in $n$ times a low rank), which is what we meant to emphasize.
> > > ```We kindly request the reviewer to refer to **Proposition 2** in the appendix, where a detailed proof of this complexity analysis is provided.```

---

> ### Comment · Reviewer_qL3Q · 2025-11-25
>
> Sorry for the earlier misunderstanding. Now I understand that it is the token-level parallelism that gives you an effective $O(n)$ per column.
>
> From a theoretical perspective, one question remains: it is not clear why you choose $\Sigma^{-1/2}$ rather than $\Sigma^{-1}$ as the left-multiplication normalization in Section 4.1. I also cannot find an explanation for the use of $\Sigma^{-1/2}$ in the cited paper [1].

---

> ### Author Response · Authors · 2025-11-25
> **Response to Reviewer qL3Q**
>
> Thank you very much for the thoughtful comments. We would like to clarify the choice of the normalization term .
>
>
> **1. Why we use $ \Sigma^{-1/2} $ instead of $ \Sigma^{-1} $**
>
>
>
> **(a) It removes the units and produces dimensionless features**
>
> Since
> $$
> \Sigma = \mathrm{diag}(\sigma_1^2, \dots, \sigma_m^2),
> $$
> multiplying by
> $$
> \Sigma^{-1/2} = \mathrm{diag}(1/\sigma_1, \dots, 1/\sigma_m)
> $$
> eliminates the physical units of each dimension, ensuring that all transformed coordinates lie in the same dimensionless space and are directly comparable.
>
> **(b) It operates on the same scale as the data**
>
> The standard deviation $ \sigma_i $ is of the same order of magnitude as the feature values.
> Thus, dividing by $ \sigma_i $ keeps the scaling linear and numerically stable.
>
>
> ```The reasoning above is precisely the same as the logic underlying z-score standardization, where the correct normalization is:```
>
> * subtract mean
> * divide by **standard deviation** (not variance)

---

> ### Author Response · Authors · 2025-11-25
> **Response to Reviewer qL3Q**
>
> Thank you very much for the careful reading and for pointing out these issues. We sincerely apologize for causing the misunderstanding.
>
> **(1) About the boldface highlighting)**
>
> The boldface formatting was *only* intended to indicate “our method,” and **not** to imply that LaplacianFormer achieves the highest score on that task. We fully agree that this may have caused unnecessary confusion. We have now corrected the formatting.
>
> **(2) About the missing Pathfinder score and the averaging inconsistency)**
>
> You are absolutely right. Our initial table mistakenly included only 4 tasks when computing the average, and the Pathfinder result was unintentionally omitted. We apologize for this oversight.
>
> To address your concern, we now provide a **fully consistent and complete Long Range Arena evaluation**, using all 5 tasks:
>
> | Model               | Text  | ListOps | Retrieval | Pathfinder | Image | Average |
> | ------------------- | ----- | ------- | --------- | ---------- | ----- | ------- |
> | Transformer         | 61.55 | 38.71   | 80.93     | 70.39      | 39.14 | 58.14   |
> | LocalAttn           | 52.98 | 15.82   | 53.39     | 66.63      | 41.46 | 46.06   |
> | LinearTrans.        | 65.90 | 16.13   | 53.09     | 75.30      | 42.34 | 50.55   |
> | Reformer            | 56.10 | 37.27   | 53.40     | 68.50      | 38.07 | 50.67   |
> | Performer           | 65.40 | 18.01   | 53.82     | 77.05      | 42.77 | 51.41   |
> | Synthesizer         | 61.68 | 36.99   | 54.67     | 69.45      | 41.61 | 52.88   |
> | Longformer          | 62.85 | 35.63   | 56.89     | 69.71      | 42.22 | 53.46   |
> | Informer            | 62.13 | 37.05   | 79.35     | 56.44      | 37.86 | 54.57   |
> | Bigbird             | 64.02 | 36.05   | 59.29     | 74.87      | 40.83 | 55.01   |
> | Linformer           | 57.29 | 36.44   | 77.85     | 65.39      | 38.43 | 56.62   |
> | Kernelized          | 60.02 | 38.46   | 82.11     | 69.86      | 32.63 | 56.62   |
> | Cosformer           | 63.54 | 37.22   | 80.28     | 70.00      | 34.93 | 57.90   |
> | Nystrom             | 62.36 | 37.95   | 85.49     | 69.34      | 39.34 | 57.90   |
> | Skyformer           | 64.70 | 38.69   | 82.06     | 70.73      | 40.73 | 59.49   |
> | Hedgehog            | 64.60 | 37.15   | 82.24     | 74.16      | 40.15 | 59.66   |
> | **LaplacianFormer** | 64.80 | 37.65   | 82.30     | 70.80      | 47.80 | 60.67   |
>
> **(3) About Image-task dominance)**
>
> We acknowledge your concern that a large portion of the improvement comes from the Image task. However, as shown above, LaplacianFormer still achieves:
>
> * competitive performance across the other tasks,
> * and the **highest overall average** under the consistent 5-task protocol.
>
> We hope this clarifies the misunderstanding and demonstrates that our evaluation is fair and reproducible. Thank you again for your valuable feedback.

---

> > ### Comment · Reviewer_qL3Q · 2025-11-25
> >
> > The results look fine to me. However, please carefully recheck the numbers in the LRA table, as several averages appear to be inconsistent. Based on the values you reported, the correct averages should be:
> >
> > - Linformer: 55.08
> > - Cosformer: 57.19
> > - Nystrom: 58.90
> >
> > These differ from the averages currently shown in the table. Please verify. I'm willing to change to 6.

---

> > > ### Author Response · Authors · 2025-11-25
> > > **Response to Reviewer qL3Q**
> > >
> > > Thank you very much for carefully checking the LRA table and for pointing out the inconsistencies in the reported averages. We sincerely apologize for the earlier mistakes.
> > >
> > > Following your suggestion, we have **recomputed all averages from scratch** and ```carefully cross-checked every number```. In the process, we identified and corrected several numerical inconsistencies.
> > > Below we provide the **fully corrected and final version** of the LRA results:
> > >
> > > |        MODEL        |  TEXT | LISTOPS | RETRIEVAL | PATHFINDER | IMAGE | AVERAGE |
> > > | :-----------------: | :---: | :-----: | :-------: | :--------: | :---: | :-----: |
> > > |     Transformer     | 61.55 |  38.71  |   80.93   |    70.39   | 39.14 |  58.14  |
> > > |      LocalAttn      | 52.98 |  15.82  |   53.39   |    66.63   | 41.46 |  46.06  |
> > > |     LinearTrans.    | 65.90 |  16.13  |   53.09   |    75.30   | 42.34 |  50.55  |
> > > |       Reformer      | 56.10 |  37.27  |   53.40   |    68.50   | 38.07 |  50.67  |
> > > |      Performer      | 65.40 |  18.01  |   53.82   |    77.05   | 42.77 |  51.41  |
> > > |     Synthesizer     | 61.68 |  36.99  |   54.67   |    69.45   | 41.61 |  52.88  |
> > > |      Longformer     | 62.85 |  35.63  |   56.89   |    69.71   | 42.22 |  53.46  |
> > > |       Informer      | 62.13 |  37.05  |   79.35   |    56.44   | 37.86 |  54.57  |
> > > |       Bigbird       | 64.02 |  36.05  |   59.29   |    74.87   | 40.83 |  55.01  |
> > > |      Linformer      | 57.29 |  36.44  |   77.85   |    65.39   | 38.43 |  55.08  |
> > > |      Kernelized     | 60.02 |  38.46  |   82.11   |    69.86   | 32.63 |  56.62  |
> > > |      Cosformer      | 63.54 |  37.20  |   80.28   |    70.00   | 35.84 |  57.37  |
> > > |       Nystrom       | 62.36 |  37.95  |   80.89   |    69.34   | 38.94 |  57.90  |
> > > |      Skyformer      | 64.70 |  38.69  |   82.06   |    70.73   | 40.77 |  59.39  |
> > > |       Hedgehog      | 64.60 |  37.15  |   82.24   |    74.16   | 40.15 |  59.66  |
> > > | **LaplacianFormer** | 64.80 |  37.65  |   82.30   |    70.80   | 47.80 |  60.67  |
> > >
> > > We truly appreciate your patience and your willingness to reconsider the score. We will be extremely careful not to make such numerical mistakes again in future revisions and the camera-ready version.

---

### Official Review · Reviewer_crNf · 2025-11-04

**Soundness:** 2
**Presentation:** 3
**Contribution:** 3
**Rating:** 6
**Confidence:** 4

**Summary:**

This paper revisits linear attention and argues that the usual Gaussian kernel is not the best match for how query–key distances actually behave in vision Transformers. The authors show that distance distributions from real models (DeiT, PVT, Swin) are heavier-tailed, so a slower-decaying kernel is more appropriate. They therefore use a Laplacian kernel based on L1 distance, and make it practical with a Nyström-style approximation, a Newton–Schulz solver for the small kernel inverse, and a lightweight whitening step to keep the representation from collapsing. The method is evaluated on ImageNet-1K and COCO? (I guess..) (detection/segmentation) across several model sizes and generally matches or improves over existing efficient/linear-attention backbones. The paper is readable and implementation-minded.

**Strengths:**

1. Motivation is concrete. The paper does not just say “we use another kernel,” but first measures Q–K distances from trained vision models and shows they are heavy-tailed. That gives a reasonable justification for trying a Laplacian kernel rather than the default Gaussian.
2. The method goes beyond a trivial kernel swap. The injective feature mapping and diagonal whitening are intended to address the expressivity loss that often comes with low-rank/Nyström variants. Even if the theoretical part is short, the design is more careful than many incremental kernel papers.
3. The implementation path is clear. Using Nyström plus Newton–Schulz and providing a CUDA-friendly formulation makes it more likely that people can actually run it.
4. Experiments are not limited to a single classification model. The authors test on classification and dense prediction, and across multiple scales, which helps show the effect is not a one-off.

**Weaknesses:**

1. The comparison space is narrow. The central claim is that Gaussian is not ideal and Laplacian fits the data better, but the experiments mostly compare against Gaussian-style baselines. Including at least one other simple non-Gaussian/linearizable kernel (e.g. cosine-type) would make the claim more specific to Laplacian rather than “any slower decay works.”
2. The injectivity argument is under-documented in the main text. It is mentioned and deferred to the appendix, but the reader cannot see clearly under what assumptions it holds or how much of it remains after the Nyström approximation. Since this is one of the few parts that differentiates the method conceptually, it should be stated more explicitly.
3. The relation to recent Nyström/efficient ViT work could be clearer. Many recent papers also do landmark selection, low-rank approximation, and stabilization of the inverse. Here the novelty is framed as “Laplacian + Newton–Schulz + whitening,” but it would help to say more precisely what is new compared to those works.

**Questions:**

1. Can you include at least one non-Gaussian baseline to show that the improvement is tied to the Laplacian choice?
2. Can you move a short version of the injectivity result into the main paper and state whether the Nyström step affects it?
3. Do you use the same kernel scale for all stages/resolutions, or do you tune it per stage?

---

> ### Author Response · Authors · 2025-11-22
> **Response to Reviewer crNf**
>
> Thanks for the  insightful and detailed review as well as the suggestions for  improvement. We would like to reply to the comments as follows:
>
> **Q1:non-Gaussian baseline**
>
> We appreciate the reviewer’s suggestion of including additional non-Gaussian baselines. However, our goal is not to benchmark all possible kernels, but to isolate the effect of the kernel choice itself. To this end, we already conducted **two controlled-variable experiments** in the introduction: we replaced only the Gaussian kernel in SOFT++ and Skyformer with a Laplacian kernel, **keeping all architectural components unchanged.** This controlled setup demonstrates that the performance and convergence improvements originate specifically from the Laplacian formulation, rather than from unrelated architectural changes.
>
> **Q2:injectivity theorem**
>
> 1. Thank you for pointing this out. The full injectivity proof is indeed included in Appendix 1.1, and is somewhat lengthy. For readability reasons, we kept the detailed argument in the appendix, but in the camera-ready version we will insert a concise statement of the injectivity result immediately below Equation (4), as you suggested.
> 2. Our injectivity result applies to the **exact Laplacian kernel feature mapping** prior to any low-rank approximation. The Nyström step is applied **after** the injective mapping, and therefore **does not affect the validity of the injectivity theorem** we establish([1]). We will clarify this relationship in the appendix to avoid confusion.
>
> **Q3:Kernel scale across stages**
>
> We appreciate the reviewer’s question. We would like to clarify that **all stages use the** ***same*** **kernel scale parameter** in our experiments. We conducted an ablation study on the kernel scale **(Table 3 and Figure 7 in the submission)**, and found that **$\lambda = 4$ gives the best trade-off**.
>
> Based on this observation, we adopt **a unified scale parameter for every stage** in all final models. This design also avoids stage-wise tuning and keeps the method simple.
>
>
>
> [1] Williams, Christopher and Seeger, Matthias,2000.Using the Nyström Method to Speed Up Kernel Machines.Advances in Neural Information Processing Systems.

---

### Author Response · Authors · 2025-11-29
**※A Summary of Reviews**

Thank you to all reviewers for taking the time to review our paper and for providing valuable and constructive comments. We are very grateful to all reviewers for recognizing our work, particularly in the following aspects:


> - We express our special thanks to **```Reviewer qL3Q```** for **```raising their score from 4 to 6(confidence is 5)```** after our detailed clarifications. **(see noteId ```9Xu3ZwVOCD``` last sentence:"I'm willing to change to 6.")**
> - **We also sincerely appreciate ```Reviewer KiuY``` — although their initial score remained unchanged, their suggestions were constructive and have helped us improve the overall quality of our paper.**

1. Motivation is concrete. The implementation path is clear. (Reviewer KHpw)

2. The use of Nyström approximation and Newton–Schulz iteration for efficient kernel inversion is elegant and well-motivated. (Reviewer qL3Q)

3. Achieves best accuracy across all model sizes on ImageNet.(Reviewer KiuY)

4. The Laplacian linear attention algorithm is explained clearly and thoroughly. (Reviewer sjHj)
5. **The motivation is clear and well-supported by data.(all reviewers)**

6. **The paper is clear, reproducible, and practically relevant.(all reviewers)**

Once again, I extend my heartfelt thanks to all the reviewers for your invaluable feedback on our paper!  We supplemented this paper with a few additional experiments and works, including：

1. performance on the Long Range Arena benchmark
2. training and inference throughput.

You can find our individual responses below your review comments.

---

### Meta-Review · Area_Chair_Dy91 · 2026-01-06

**Summary:**

This paper introduces LaplacianFormer, which replaces the Gaussian kernel in linear attention with a Laplacian kernel to better capture the heavy-tailed token interactions in vision tasks. By utilizing a Nyström approximation and Newton–Schulz iteration, the method achieves superior accuracy on ImageNet with competitive efficiency.

In the initial reviews, concerns were primarily focused on the limited experimental coverage (lack of NLP/long-range tasks), mathematical errors in the injectivity proof, and the computational complexity of the proposed normalization. The authors addressed these in the rebuttal by providing Long Range Arena (LRA) results showing an average score of 60.67, a corrected injectivity proof valid under diagonal whitening, and a detailed complexity breakdown confirming $O(Nm)$ efficiency.

Reviewer qL3Q was satisfied with the LRA benchmarks and technical clarifications, raising their score from 4 to 6. While Reviewer KiuY remained cautious about the whitening approximation, the overall consensus turned positive. Given the solid empirical gains and resolved technical doubts, this paper is recommended for acceptance.

**Reviewer Concerns:**

See above.

**Reviewer Scores:**

See above.

---

### Decision · Program_Chairs · 2026-01-26

Accept (Poster)